

# Maximizing Ozone Signals Among Chemical, Meteorological, and Climatological Variability

Benjamin Brown-Steiner[,1,2,3], Noelle E. Selin[2,4,5], Ronald G. Prinn[1,2,5], Erwan Monier[1,2], Simone Tilmes[6], Louisa Emmons[6], Fernando Garcia-Menendez[7]

*Correspondence to: Benjamin Brown-Steiner* (bbrownst@aer.com)

1. Center for Global Change Science, Massachusetts Institute of Technology, 7 Massachusetts Ave, Cambridge, MA 02139
2. Joint Program on the Science and Policy of Global Change, Massachusetts Institute of Technology, 7 Massachusetts Ave, Cambridge, MA 02139
3. Now at Atmospheric and Environmental Research, Lexington, Massachusetts, 02421
4. Institute for Data, Systems, and Society, Massachusetts Institute of Technology, 7 Massachusetts Ave, Cambridge, MA 02139
5. Department of Earth, Atmospheric, and Planetary Sciences, Massachusetts Institute of Technology, 7 Massachusetts Ave, Cambridge, MA 02139
6. Atmospheric Chemistry Observations and Modeling Lab, National Center for Atmospheric Research, 3450 Mitchell Lane, Boulder, CO 80301
7. Department of Civil, Construction, and Environmental Engineering, North Carolina State University, Raleigh, NC 27695



**Abstract**
The detection of meteorological, chemical, or other signals in modeled or observed air quality
data – such as an estimate of a temporal trend in surface ozone data, or an estimate of the mean
ozone of a particular region during a particular season – is a critical component of modern
atmospheric chemistry. However, the magnitude of a surface air quality signal is generally small
compared to the magnitude of the underlying chemical and meteorological variabilities that exist
both in space and in time. This can present difficulties for both policy-makers and researchers as
they attempt to identify the influence or 'signal' of climate trends (e.g. any pauses in warming
trends), the impact of enacted emission reductions policies (e.g. United States $NO_x$ State
Implementation Plans), or an estimate of the mean state of highly variable data (e.g. summertime
ozone over the Northeastern United States).  Here we examine the scale-dependence of the
variability of simulated and observed surface ozone data within the United States and the
likelihood that a particular choice of temporal or spatial averaging scales produce a misleading
estimate of a particular ozone signal. Our main objective is to develop strategies that reduce the
likelihood of overconfidence in simulated ozone estimates. We find that while increasing the
extent of both temporal and spatial averaging can enhance signal detection capabilities by
reducing the 'noise' from variability, a strategic combination of particular temporal and spatial
averaging scales can maximize signal detection capabilities over much of the Continental US.
We recommend temporal averaging of at least 10 - 15 years combined with regional spatial
averaging over several hundred kilometer spatial scales. These results are consistent between
simulated and observed data, and within a single model with different sets of parameters. The
strategies selected in this study are not limited to surface ozone data, and could potentially
maximize signal detection capabilities within a broad array of climate and chemical observations
or model output.





**Copyright Statement**
• Authors retain the copyright of the article. Regarding copyright transfers please see
below.
• Authors grant Copernicus Publications an irrevocable non-exclusive license to publish
the article electronically and in print format and to identify itself as the original publisher.
• Authors grant Copernicus Publications commercial rights to produce hardcopy volumes
of the journal for sale to libraries and individuals.
• Authors grant any third party the right to use the article freely as long as its original
authors and citation details are identified.
• The article is distributed under the Creative Commons Attribution 4.0 License. Unless
otherwise stated, associated published material is distributed under the same license.



## 1 Introduction


The capability to detect air quality signals – be they meteorological, chemical, or of some
other type – is a fundamental component of modern climate science and atmospheric chemistry.
The debate over the existence or length of a global warming hiatus (Lewandowski et al., 2015;
Roberts et al., 2015; Medhaug et al., 2017) and research examining the time of emergence of
climatological (Hawkins and Sutton, 2012; Elía et al., 2013; Schurer et al., 2013), meteorological
(Giorgi and Bi, 2009; King et al., 2015), chemical (Barnes et al., 2016; Garcia-Menendez et al.,
2017), and other sectoral signals (e.g. Monier et al., 2016) embody an accumulation of
techniques and strategies for filtering noise (due to natural variability) and maximizing the
capability to detect statistically significant signals and trends in noisy data. It is well established
that temporal averaging (e.g. Lewandowski et al., 2015) and spatial averaging (e.g. Frost et al.,
2006; Hawkins and Sutton, 2012; Barnes et al., 2016) can enhance signal detection capabilities
in atmospheric data. Here we extend this research by quantifying the impact of both spatial and
temporal averaging – individually and in combination – of surface ozone on the magnitude of the
calculated variability, which is largely driven by the influence of meteorological variability on
the atmospheric chemistry (e.g. Jacob and Winner, 2009). We offer recommendations for
strategically averaging in space and time to maximize signal detection capabilities. In particular,
we examine estimates of mean ozone and of the ozone variability that results from meteorology,
although our approach can be generalized to other air quality applications.
For observed ozone data, strategies for reducing spatial and temporal noise are limited: a
longer time series is needed, more observations need to be made, or the spatial region over which
the ozone observations are being averaged over needs to be enlarged. For surface ozone
estimates using models, however, there exist a variety of strategies for reducing the noise (due to
chemical and meteorological variability) relative to the strength of the signal, although they
cluster into three main types. The first strategy is to average or combine multiple runs of
structurally different models under the assumption that errors, biases, and uncertainties within
the individual models are reduced and the multi-model or multi-dataset mean is a best estimate
of the actual, aggregated ozone field. This is most notably done with multi-model ensembles
within the ACCMIP framework (Lamarque et al., 2013; Young et al., 2013; Stevenson et al.,
2013), and this approach tends to assume that all members in the ensemble are independent and
equally skillful. This assumption, however, may result in a loss of some valuable information




(Knutti, 2010). Another form of this strategy is to run multiple model runs within a single model,
but under different initial conditions or sets of parametric assumptions (e.g. Deser et al, 2010;
Monier et al., 2013, 2015; Kay et al., 2015; Garcia-Menendez et al., 2015, 2017). This approach
cannot address structural uncertainties between models, but is capable of identifying parametric
uncertainties within a single model.

The second strategy to reduce ozone variability is to expand the temporal averaging window,

which can influence the interpretation of the determined ozone value (e.g. Brown-Steiner et al.,
2015). The Environmental Protection Agency (EPA) National Ambient Air Quality Standard
(NAAQS) for ozone (US EPA, 2015) explicitly takes this into account, both in the length of the
averaging period (daily maximum 8-hour average) and the selection criteria for the standard
(fourth-highest over the previous 3 years). The calculated ozone variability can be further
reduced by utilizing even longer averaging periods, such as monthly (e.g. Rasmussen et al.,
2012), seasonal (e.g. Fiore et al., 2014; Barnes et al., 2016), annual, or decadal mean values (e.g.
Garcia-Menendez et al., 2017). This strategy is analogous to the averaging of meteorological
data to derive a climate signal, and just as Lewandowsky et al. (2015) recommend averaging 17
or more years in order to achieve climatological estimates of temperature trends, there is a
growing body of literature recommending averaging short time scale chemical variability (what
could be called chemical weather, see Lawrence, 2005) for 15 or more years (e.g. Garcia-
Menendez et al, 2017) in order to achieve an estimate of the what could be called the chemical
climate (see Möller, 2010).

The third strategy to reduce ozone variability is to average surface ozone values over larger

spatial regions, and while there is a significant body of literature discussing the capability and
interpretation of coarse resolution model representations of the sub-grid scale heterogeneity
(Pyle and Zavody, 1990; Searle et al., 1998, Wild et al., 2006), there are few that strategically
expand the spatial scale over which averaging is applied in order to maximize signal detection
capabilities. This strategy has been applied in other fields of the atmospheric sciences as well as
for general gridded datasets (e.g. Pogson and Smith, 2015), and spatial averaging has been
suggested as a means of reducing temperature variability and smoothing biases at the smallest
spatial scales within a single model run (Räisänen and Ylhäsi, 2011). This "scale problem" has
also been noted as an important consideration when analyzing aerosol indirect effects



(McComiskey and Feingold, 2012) and for the detection and attribution of extreme weather
events (Angélil et al., 2017).
Our objective in this study is to provide a framework for selecting spatial and temporal
averaging scales that limits the likelihood of over-confidence in an estimate of surface ozone that
arises from meteorological variability. This type of framework can be useful from two different
research perspectives. The first research perspective has a priori an ozone estimate (either
observed or modeled) at a certain spatial and temporal scale (e.g. a 3-year simulation of surface
ozone over the Northeastern US) and wants to quantify the likelihood that this estimate is
representative of the long-term ozone behavior (rather than overly sensitive to meteorological
variability of that particular 3-year period). Since ozone is strongly influenced by natural
fluctuations in meteorology (Jacob and Winner, 2009; Jhun et al., 2015) and since extremes in
surface ozone and temperature tend to co-occur (Schnell and Prather, 2017), atypically hot or
cold periods can strongly influence ozone behavior over short time scales.
The second research perspective is to identify an ozone signal of a certain magnitude (or
threshold) and needs to decide what spatial and temporal averaging scales are needed to best
identify that signal. The ozone signal could be large (e.g. determining the effectiveness or
compliance with a 5 ppbv incremental reduction of the EPA NAAQS for ozone (US EPA, 2015))
or small (e.g. identifying annual ozone trends within the US, which Cooper et al. (2012) show
can be on the order of $0.10 - 0.45$ ppbv), and can be highly sensitivity to spatial and temporal
heterogeneity and meteorological variability. Barnes et al. (2016) found that surface ozone trends
over 20-year periods can vary by $\pm 2$ ppbv due solely to climate variability, while interannual
variability can be on the order of $\pm 15$ ppbv (Fiore et al., 2003; Tilmes et al., 2012; Line et al.,
2014) and day-to-day variability can be even larger, extending regularly from near-background
levels of $40 - 50$ ppbv up to 100 ppbv during the summertime (Fiore et al., 2014).
In this study, we quantify the impact of both temporal and spatial averaging on the calculated
ozone variability – due solely to meteorological variability – in order to maximize the capability
to detect trends. We use simulated ozone (with the Community Atmosphere Model with
Chemistry, CAM-chem) and observational data (with the EPA's Clean Air Status and Trends
Network, CASTNET) within the United States in order to answer the following four questions:
(1) Within a given dataset (model or observations), with both spatial and temporal coverage,
what is the magnitude of the ozone variability due to meteorology at the smallest scale, and how



does spatial and temporal averaging reduce this variability? (2) Are there combinations of
temporal and spatial averaging scales that maximize the signal detection capability for surface
ozone data? (3) How sensitive are the above strategies to different configurations (i.e. emissions,
meteorology, and climate) of the CAM-chem modeling framework? And (4) How could they be
applied to other datasets (chemical, meteorological, or climatological)? We limit our focus to
spatial scales within the United States as it has high spatial and temporal variability and
numerous observations, and since averaging over larger regions (e.g. the Northern Hemisphere,
or the globe) would produce a smaller calculated variability.
In Section 2, we describe the CAM-chem model and our simulations, as well as the
CASTNET observational database and the regional definitions used throughout this paper. In
Section 3 we quantify the temporal and spatial variability of surface ozone, show how temporal
and spatial averaging reduces the calculated ozone variability, and demonstrate the spatial
heterogeneity of the calculated ozone variability. In Section 4, we discuss the potential strategies
that could be used to maximize ozone trend detection due to meteorological variability, explore
uncertainties, and make recommendations for future research.

**2 Methods**


We examine both present-day (one simulation and one observed dataset) and future (two
simulations) surface ozone in this study. For present-day analysis, we simulate surface ozone
using CAM-chem, a component of the Community Earth System Model (CESM) and available
observations within the US from the EPA CASTNET database. For future analysis, and in order
to examine the potential for patterns of variability to change in the future, we utilize two existing
simulations of CAM-chem conducted by Garcia-Menendez et al. (2017). Much of this analysis is
conducted using the R language (R-Project, www.r-project.org). Here we summarize each of the
three datasets and our approach to our analysis in Section 3.

**2.1 CAM-chem**

The present-day simulation (MOZ_2000) was conducted using CAM-chem model
version 1.2.2, with the CAM4 atmospheric component (Tilmes et al., 2015; 2016). The model
has been used extensively for a wide range of atmospheric chemistry research and included in
the Atmospheric Chemistry and Climate Model Intercomparison Project (ACCMIP, Lamarque et



al., 2012; Young et al., 2012 and references therein). We conduct our simulations using the
MOZART-4 chemical mechanism (Emmons et al., 2010) with offline forced meteorology from
the Modern-Era Retrospective analysis for Research and Applications (MERRA) reanalysis
product (Rienecker et al., 2011) for 26 meteorological years (1990 – 2015). This simulation has
56 vertical levels – adopted from MERRA meteorology – and 96 latitudinal and 144 longitudinal
grid cells. We aim to isolate the variability to the meteorologically-driven impact on atmospheric
chemistry so we repeat year-2000 anthropogenic emissions from the ACCMIP (Atmospheric
Chemistry and Climate Model Intercomparison Project) inventory (Lamarque et al., 2012) and
all non-biogenic emissions for all meteorological years, and include specified long-lived
stratospheric species ($O_3$, $NO_x$, $HNO_3$, $N_2O$, $N_2O_5$) as in MOZART-4 (Emmons et al., 2010), an
online biogenic emissions model MEGAN (Guenther et al., 2012), and forced sea ice and sea
surface temperatures to year 2000 historical conditions. Like many state-of-the-art chemical
tracer models, the CAM-chem exhibits some biases, most notably for our purposes a high bias in
simulated surface ozone in the Eastern US (e.g. Lamarque et al., 2012; Brown-Steiner et al.,
2015; Travis et al., 2016; Barnes et al., 2016). Recent efforts have been successful in partially
reducing these biases (e.g. Sun et al., 2017).

We also include two reference simulations of the future, MOZ_2050 and MOZ_2100

(simulating the meteorological years 2035 – 2065 and 2085 – 2115, respectively) using the
CESM CAM-chem simulations described in detail by Garcia-Menendez et al. (2017) with one
set of initial condition data, and a climate sensitivity of 3.0 ˚C.  Compared to the present-day
simulations, these future simulations have several parametric differences: the model version is
1.1.2, the atmospheric component is CAM3, the emissions (which are held constant at year-2000
levels) are from the Precursors of Ozone and their Effects in the Troposphere database (see
Garcia-Menendez et al., 2017), and the meteorology is derived from a linkage between the
Massachusetts Institute of Technology Integrated Global System Model (MIT IGSM) and the
CESM CAM model (Monier et al., 2013), and as such has 26 vertical levels. For a full
description of these simulations, see Garcia-Menendez et al. (2017).

**2.2 CASTNET**

The observational database comes from the EPA Clean Air Status and Trends Network

(CASTNET), which has more than 90 surface observational sites within the United States and



has been collecting hourly surface meteorological and chemical data since 1990 (US EPA, 2016
and https://www.epa.gov/castnet). We collected data from all sites that reported complete ozone
data from each year and removed data that was marked invalid within the downloaded EPA files.
The number of sites that matched these criteria varied from year to year, but generally we have
between 55 and 94 sites throughout the 1991 – 2014 period. The CASTNET observational
network is located primarily in rural sites, and thus is considered to be a reasonable comparison
to coarse grid cell model output. Since a notable trend in observed ozone data exists, especially
in the Northeastern US (Frost et al., 2006), and since the simulations have no change in
anthropogenic emissions, and thus no ozone trend, we detrended the CASTNET data for each of
the four averaging regions (described below) using a simple linear regression.

**2.3 Telescoping Regional Definitions**

In order to isolate the impact of the size of the spatial scale over which ozone data is

averaged, we analyze ozone data at different spatial scales. The largest region considered is the
entire Continental US, while the smallest regions considered are at the individual grid cell level
of the CESM CAM-chem model (1.9˚x2.5˚ latitude/longitude). We focus on the US since there
are CASTNET observations that provide adequate coverage in both space and time, and since the
US has significant temporal and spatial variability. Data and statistics for the other regions (i.e.
the Midwestern and Southeastern US) are included in the Supplemental Material, but do not alter
the conclusions we draw from the Northeastern US. For CESM CAM-chem data, we averaged
all grid cells within each region, while for the CASTNET data we first average sites within each
corresponding CESM CAM-chem grid cell, and then averaged these data together. These
telescoping regions are shown in Figure 1.

**3 Results**

Here we examine the spatial and temporal behavior of MOZ_2000, MOZ_2050, and

MOZ_2100 and compare MOZ_2000 to present-day CASTNET observations. We introduce the
moving temporal averaging windows, explore possible thresholds of acceptable error or signal
strength, and examine the influence of expanding spatial averaging regions. Finally, we combine
these temporal and spatial averaging techniques into a single framework.





**3.1 Spatial and Temporal Comparisons**
Figure 2 plots the averaged spatial distribution of the daily maximum 8-hour ozone
average (DM8H $O_3$) for summertime (JJA) days for 1990-2015 for the present-day MOZART
simulation, MOZ_2000 (Figure 2a) and for the year 2000 for CASTNET data (Figure 2b). The
well-known high ozone bias in the Eastern US (e.g. Lamarque et al., 2012; Travis et al., 2016;
Barnes et al., 2016) is apparent, but otherwise the spatial variability over the entire Continental
US is well captured. While we do examine the magnitude of surface ozone in this paper, most of
our analysis is focused on the variability around the mean value (the anomaly), and as we show
below, the CASTNET observations and CESM results are largely consistent in their
representation of ozone variability. The standard deviation of DM8H $O_3$ is large over the Eastern
US and the Pacific Coast, with peak values of ± 25 ppbv over the highly populated Atlantic
Coast (Figure 2c). The variability (defined as the standard deviation divided by the mean,
expressed as a percentage) is lowest over the Western US (~ 15%), only slightly higher over the
Eastern US (up to 25%), and highest (up to 50%) over the coastal regions (Figure 2d). The future
simulations, MOZ_2050 and MOZ_2100 (Figure 2e and 2f, respectively), although run with
different parametric settings than MOZ_2000 (see Section 2), simulate a similar spatial
distribution of surface ozone, although under the warmer simulated climate of 2050 and 2100.
These future simulations have a similar spatial pattern to the present-day simulation (Figure 2a),
with high ozone levels in the Eastern US that increases from 2050 to 2100 (see Garcia-Menendez
et al. (2017) for more details).
Figure 3 compares boxplots over the four telescoping regions (Figure 1) for MOZ_2000,
the CASTNET data, the detrended CASTNET data, and for the single year 2000 for the
CASTNET data (Figures 3a-d), and Table 1 summarizes relevant statistics. In order to compare
CASTNET ozone to the simulated ozone, which do not have a trend over time, we detrend the
CASTNET data in order to remove the impact of any temporal trends (e.g. $NO_x$ emissions
reductions) on ozone. The Northeastern US ozone bias is apparent at the smaller spatial scales
(Figures 3c,d) and is less apparent when averaging over larger regions (Figures 3a,b). Figure 3e
compares the year-to-year boxplots of the JJA DM8H $O_3$ for the MOZ_2000 and the detrended
CASTNET data, and demonstrates the variability both in the median and spread of the ozone
values in both the modeled and simulated data. While the MOZ_2000 ozone is generally higher
than the CASTNET data, there are years in which the CASTNET data has higher ozone





extremes. The red box plot in Figure 3e, which corresponds to the red box plot in Figure 3b,
indicates that the year 2000 was an anomalously low year for observed ozone, although not the
lowest.
While all the CESM CAM-chem simulations have high ozone biases in the Northeastern
US (Figures 2 and 3, Table 1), their capability to simulate ozone variability is consistent with the
available observations (for present day) and for expectations of ozone variability changes in the
future (for MOZ_2050 and MOZ_2100). Here we examine the variability defined as the standard
deviation divided by the mean (expressed as a percent), instead of the standard deviation alone,
in order to account for the model biases in the magnitude of the simulated ozone. It is clear that
variability increases when the size of the averaging region decreases, a fact that is well noted in
the literature, as in Hawkins and Sutton (2012) for climate variables and Barnes et al. (2016) for
ozone. As can be seen in in Table 1, the CASTNET variability increases as the spatial scale
decreases (10%, 13%, 16%, and 20% for our telescoping regions), and MOZ_2000 largely
captures this trend (5%, 10%, 15%, and 15%). This increase in ozone variability with decreasing
spatial scale is maintained in the future simulations (6%, 10%, 16%, and 21% for MOZ_2050
and 7%, 12%, 17%, and 20% for MOZ_2100). Table S1 contains statistics for the other
telescoping regions.
**3.2 Variability, Averaging Windows, and Thresholds**
As we aim to quantify the potential tradeoffs that result from a particular choice of
temporal and spatial scales on the assessment of ozone variability within the US, we represent
the spatial scale by applying the telescoping regions (see Figure 1) and we represent the temporal
scale through the use of moving averaging windows that range from 1 day up to the full 26 years
for the CESM data (1990-2015), the full 24 years for the detrended CASTNET data (1991 –
2014), and the 30 years available from the future scenarios of Garcia-Menendez et al. (2017).
Each averaging window, therefore, can be considered to be a "sample" of possible realizations of
meteorology. For instance, a selection of an averaging window of 1 year has 26 possible slices
within the 1990 – 2015 MOZ_2000 data, while a selection of an averaging window of 10 years
has 17 possible slices within the CESM data (N = # years – length of window +1). In this study,
we consider all realizations to be equally likely and compare them to each other and to the long-
term trend. However, if we were only able to simulate 5 years, we would not be able to compare




to the long-term trend, and so be unable to completely quantify the likelihood of error in the
context of the long-term behavior. We frame much of the following analysis from the
perspective of limited simulation length in order to approximate the question that decision-
makers and modelers face when constrained by limited computational capabilities or available
data: what's the likelihood that a particular estimate (of both the mean and the variability) is not
a true representation of the true mean and variability, but rather a product of the particular choice
of spatial and temporal scale?

Figure 4 presents this likelihood by plotting all possible estimates of DM8H $O_3$ (as

anomalies from the long-term mean) over all possible selections of averaging window (from 1
day up to the complete time series) for our telescoping regions. The semi-cyclical and highly
auto-correlated nature of surface ozone is apparent at all spatial scales, with alternating cycles of
anomalously high and low ozone. The temporal impact of anomalous ozone events is indicated
by the vertical and right-leaning diagonal striations, which show that anomalous ozone events
can impact estimates of ozone values within averaging windows up to 15 or 20 years. Figure 4
demonstrates how small-scale anomalously high or low ozone values (that come only from
meteorological variability) can impact temporal averages of 5, 10, or even 20 years. For instance,
a selected 5-year averaging window within the MOZ_2000 simulation averaged over the
Northeastern US could be 2.5 ppbv higher or lower than the 25-year mean value of 74 ppbv, a
difference of 7%. Horizontal lines in Figure 4 mark the length of averaging windows that are
needed to ensure that ozone variability does not exceed a given threshold (5, 1, and 0.5 ppbv for
solid, dashed, and dotted lines respectively). This difference is larger within smaller regions and
at the shorter selections of the averaging window. While the high and low ozone anomalies differ
in time between CASTNET, MOZ_2000, MOZ_2050, and MOZ_2100 in Figure 4, the impact of
spatial and temporal averaging is consistent.

We also quantify this variability in Supplemental Figures S1 and S2, which plots the

likelihood (as a percentage) that a particular selection of spatial (rows) and temporal (x-axis)
scale estimates ozone values that exceed a particular threshold (colored lines) away from the true
mean value. For instance, if we are interested in characterizing ozone behavior (e.g. estimating a
trend, or the mean value) in the Northeastern US, but were limited to a 5-year simulation, there is
more than a 50% likelihood that the simulated ozone is 1 ppbv away from the 26-year mean, and
an 80% likelihood that the discrepency is greater than 0.5 ppbv. However, these data indicate





that there is a virtual certainty that the estimate will be within 2.5 ppbv of the true mean value.
We should note that, at the grid-cell level and within a 10-year period, the surface ozone
variability can exceed 1 ppbv but is unlikely to exceed 2.5 ppbv (Figure 4), and that a 20-year
trend is very likely to be able to identify significant ozone signals among the impact of
meteorological variability on atmospheric chemistry. Our results also align with the results from
Garcia-Menendez et al. (2017), which recommended that simulations need to be at least 15 years
long to identify anthropogenically-forced ozone signals on the order of 1 ppbv.

Figures 4 and Supplemental Figures S1 and S2 compare the CASTNET observations to

the three CESM CAM-chem simulations, and while there are minor differences, there are broad
features that are consistent. First, using longer temporal averaging windows reduces the
influence of small-scale ozone variability at all spatial scales, and depending on the acceptable
threshold, one can select a temporal scale that effectively reduces the likelihood of exceeding
that threshold to zero. Second, larger spatial scales also reduce this likelihood of exceeding a
given threshold, but not as effectively as longer temporal scales. Finally, the impact of both
temporal and spatial averaging on ozone variability is largely consistent for the CASTNET
observations and for all three CESM CAM-chem simulations.

**3.3 Selection of Temporal Averaging Scales**

Figure 5 extends this analysis to examine the spatial heterogeneity of this likelihood of

exceeding particular thresholds at the grid cell level. Here we plot four thresholds (0.5, 1, 2.5,
and 5 ppbv) and four averaging windows (1, 5, 10, and 20 years) for the MOZ_2000 simulation.
Ozone variability is highest in the Eastern US. At the grid-cell level, there are two strategies for
filtering out the noise associated with natural meteorological variability (and thus enhancing
signal detection capabilities): either average over longer periods, or increase the threshold. For
these data, it is virtually certain that any 20-year average will be within 5 ppbv of a full 25-year
mean value (which itself may not be an accurate representation of a longer simulation), and
virtually certain that any 1-year average will be at least 0.5 ppbv away from the mean.

Figure 6 and Supplemental Figure S3 compare the MOZ_2000, MOZ_2050, and

MOZ_2100 simulations by selecting one column (the 5-year averaging window) and one row
(the 1 ppbv ozone threshold) from Figure 6 for MOZ_2000 to equivalent plots for MOZ_2050
and MOZ_2100. Interpreting Figures 7 and Supplemental Figure S3 give largely consistent



interpretations than the analysis above. Namely, that at the grid-scale level, increasing the
temporal averaging window (Figure 6) or increasing the acceptable ozone threshold
(Supplemental Figure S3) are effective at reducing the impact of the meteorological variability
on estimates of the ozone signal. Short windows (or smaller thresholds) are needed in the
Western US than in the Eastern US, and grid-cells over coastal and highly populated regions tend
to need longer windows (or higher thresholds). Finally, the 1 ppbv threshold and the 5-year
averaging window plots (in either Figure 5 or Supplemental Figure S3) indicate that the spatial
distribution and location of the peak variability may shift into the future, although this may be
due to parametric differences between MOZ_2000, MOZ_2050, and MOZ_2100. Future
simulations will be needed to check this shift in peak ozone variability.

**382    3.4 Selection of Spatial Averaging Scales**

We examine the impact of increasing the spatial averaging region (Figure 7) at four

different temporal averaging windows (1, 5, 10, and 20 years) and for the smallest ozone
threshold from the previous section (0.5 ppbv). It is evident that at all temporal averaging
windows, expanding the number of surrounding grid cells that are averaged together consistently
decreases the likelihood of exceeding the 0.5 ppbv threshold, although these reductions are
relatively small at the 1-year window, especially over the Eastern U.S. While increasing the
spatial averaging from a single grid-cell up to include the surrounding 81 grid cells (bottom row
in Figure 7) manages to essentially smooth away much of the spatial heterogeneity in surface
ozone (by moving down any column in Figure 7), it does not eliminate the likelihood of
exceeding the 0.5 ppbv threshold over much of the Eastern U.S. For instance, even at a 20-year
averaging window, and by averaging together the surrounding 81 grid-cells over locations in the
Eastern U.S., there is still a 20-70% likelihood of exceeding the 0.5 ppbv threshold due to the
small-scale impact of the meteorological variability on atmospheric chemistry.

**397    3.5 Combination of Spatial and Averaging Scales**

We now examine the combined impact of temporal and spatial averaging on reducing the

influence of small-scale ozone variability in order to enhance ozone signal detection capabilities.
Table S2 summarizes our analysis by dividing the likelihood of the ozone variability estimates
exceeding selected thresholds away from the long-term mean into four categories: (1) the length



of the averaging window over which ozone is averaged (columns); (2) the magnitude of the
ozone threshold of interest (rows); (3) the observed (CASTNET) and modeled (MOZ_2000,
MOZ_2050, and MOZ_2100) ozone data (sub-columns); and (4) the size of the spatial extent
over which ozone is averaged (sub-rows). A graphical representation consistent with the data
presented in Table S2 is plotted in Figure 8 for the Continental US average and for three grid
cells that represent various cases. In each plot in Figure 8, by moving along columns from left to
right, we can see the influence of increasing the size of the temporal averaging window, and by
moving along rows (from the bottom to the top), we can see the influence of increasing the
spatial averaging scale. By taking in the entire plot as a whole, we can get a feel for the
combined influence of both temporal and spatial averaging. Supplemental Figure S4 contains a
plot for each grid cell in the Continental US.
On average within the Continental US, both temporal and spatial averaging are effective
at reducing the calculated DM8H $O_3$ anomaly, although temporal averaging is more effective
(Figure 8a). There are many grid cells in the Eastern and Western US coasts (Figure 8b,
Supplemental Figure S4), where both spatial and temporal averaging are effective, but their
combined usage is especially effective. There are also many grid cells where temporal averaging
is effective, but spatial averaging is barely effective, or not effective at all (Figure 8c and
Supplemental Figure S4). Finally, there are some grid cells, particularly in the Central US
(Figure 8d and Supplemental Figure S4), where spatial averaging over smaller regions is
effective, but spatial averaging of larger regions actually increases the calculated DM8H $O_3$
anomaly by including surrounding grid cells that have higher variability.

**4 Discussion**
We now return to the original three research questions posed in Section 1. First, what is
the magnitude of ozone variability due to meteorology alone at the smallest scale, and what is the
impact of increasing the scale of temporal and spatial averaging? In both observed and modeled
DM8H $O_3$ surface data, the small-scale variability driven solely by the meteorological variability
impact on atmospheric chemistry (expressed as the standard deviation as a percentage of the
mean) can exceed 20% (Table 1, Figure 2d). The chemical variability examined here is the result
of fluctuations in meteorology, which itself results from larger-scale climatological drivers.
While variability in emissions also influences atmospheric chemistry, our analysis has removed





the influence of emissions variability and isolated the variability due to meteorology. There is
high temporal and spatial heterogeneity of surface ozone (Figure 2d), with the lowest values
found in the Western US (< 10%), higher values found in the Eastern US (up to 20%), and the
highest values over coastal or heavily populated regions (up to 30%). Averaging over longer
temporal scales (by increasing the averaging window) and over larger spatial scales (by
expanding the averaging region) can reduce the magnitude of the calculated variability, with
temporal averaging proving to be more effective than spatial averaging in most cases (Figure 8).
In this study, we performed simple spatial averaging, but there are other methodologies for
smoothing two-dimensional signals (e.g. Räisänen et al., 2011; Pogson and Smith, 2015) that
could potentially increase signal detection capabilities.

Second, are there combinations of temporal and spatial averaging that maximize the

filtration of calculated ozone variability, and thus maximize the potential for signal detection?
Figure 8 (and Supplemental Figure S4) demonstrate clearly that there are cases in which the
combined usage of temporal and spatial averaging can reduce the calculated variability better
than either strategy alone (see Figure 8b), although there are many regions within the Eastern US
in which spatial averaging has little to no impact on reducing the calculated variability (Figure
8c) or even results in an increase in the calculated variability (Figure 8d). There are no such
cases (see Supplemental Figure S4) in which expanding the temporal averaging scale increases
the calculated ozone variability. This could potentially enable region-specific averaging
strategies that help decision-makers identify and meet regional air quality objectives.

Third, are these results dependent on the particular parameterizations of the CESM

CAM-chem model, are they consistent with the available CASTNET observations? The three
CESM CAM-chem simulations exhibited consistent representations of ozone variability,
consistent with our understanding of future changes to the climate (and meteorology) and the
resulting impact on atmospheric chemistry (Table 1, Figure 4, S1, and S2). Compared to the
CASTNET observations (which we detrended to remove the influence of changing precursor
emissions), the present-day simulation (MOZ_2000) exhibited a high ozone bias in the Eastern
US (which is also evident in the future simulations, MOZ_2050 and MOZ_2100), while the
representation of the ozone variability is comparable (Table 1).

Finally, how may these strategies be applied to other datasets, be they chemical,

meteorological, or climatological?  Much of this analysis could be applied to any dataset that has





spatial and temporal coverage, as long as some set of acceptable thresholds is provided. While
our time step in this analysis is daily (given the DM8H $O_3$ metric), and applied only to
summertime (JJA) days, any time step (i.e. hourly, monthly, annual, decadal) could be utilized as
long as cyclical trends (e.g. diurnal or seasonal cycles) are removed. Indeed, the sliding-scale
presentation in Figure 8 and Supplemental Figure S4 can specifically be utilized to identify
particular spatial and temporal scales that are sufficient to identify signals at particular thresholds
and to identify particular geographic regions that are best suited to identify a given signal. For
example, Sofen et al. (2016) identified regions across the globe where additional observations
would be particularly suited to improve our understanding of surface ozone behavior, and our
analysis could potentially be used to identify particular temporal and spatial averaging scales that
could further maximize the capability for trend detection. In particular, Sofen et al. (2016) noted
that the peak in the power spectrum of the El Niño-Southern Oscillation (ENSO) on surface
ozone is at the 3.8 year time scale, and that within some regions within the US, the amplitude of
the ENSO influence on surface ozone approached 0.5 ppbv (and up to 1.1 ppbv globally). Our
analysis shows that there are no grid cells within the Continental US where a 0.5 ppbv signal can
be identified at the 5-year (or shorter) temporal averaging scale (Supplemental Figure S4), but
that there are many regions – especially within the Western US – in which even a modest amount
of spatial averaging can identify surface ozone signals below the 1 ppbv level with a 5-year or
shorter averaging window. The type of sliding-scale analysis – in which spatial and temporal
averaging are utilized individually and in combination – as presented in Figure 8 and
Supplemental Figure S4 could readily be applied to a wide range of atmospheric (and other)
topics to aid in the capability to identify signals that exist both in space and in time. In particular,
low-frequency oscillations (e.g. ENSO, and others) and other forms of internally or externally
forced trends (e.g. anthropogenic and natural changes in emissions) are readily adaptable to this
type of analysis.

Finally, we did not quantify statistical significance (as in Lewandowski et al., 2015) as

our goals were to understand the general nature of ozone variability at all scales and for all signal
strengths. Statistical significance testing (and other statistical techniques) can certainly provide
additional information as to the strengths of ozone signals within the underlying variability, and
can be used to extend these results in a case-by-case manner, but we leave this testing to future
studies that can focus on particular air quality objectives at particular temporal and spatial scales.




**5 Conclusions**


We quantified the impact of spatial and temporal averaging at different scales – both
individually and combined – on estimates of surface ozone variability and the resulting
likelihood of over-confidence in estimates of chemical signals over the United States using
CASTNET observations and the CESM CAM-chem model. We simulate three multi-decadal
time periods, each with constant surface emissions, and find that this analysis is consistent across
our simulated time periods, and that our results are not sensitive to particular configurations
parametric choices within the CESM CAM-chem (i.e. emissions, meteorology, and climate). We
also provide a conceptual framework for gaining understanding of the influence of spatial and
temporal averaging that may be adapted to a wide range of atmospheric and surface phenomena,
provided sufficient spatial and temporal coverage. Here we focus on surface ozone, a highly
variable (in both space and time) atmospheric constituent with severe human health impacts and
implications for planetary climate, which is the focus of many local, regional, and national
policies. However, the resultant magnitude of these changes and trends are small compared to the
magnitude of the day-to-day ozone variability, and detecting these changes and trends can be
challenging. Our analysis and conceptual framework allow for a selection of spatial and temporal
averaging scales that can aid in this signal detection.
In order to quantify the impact of spatial and temporal averaging on ozone variability, we
start by selecting four telescoping spatial regions (the Continental US, the Eastern US, the
Northeastern US, and a single grid cell within the Northeastern US) and examine all possible
choices for averaging windows (ranging from daily to multi-decadal windows), although we
focused primarily on averaging windows of 1, 5, 10, and 20 years. We find that – consistent with
previous studies – ozone variability is largest at the smallest scales, and is frequently on the order
of $\pm 10 - 20$ ppbv, or which is roughly 15-20% of the mean ozone signal. In order to minimize
the chemical noise that results from meteorological variability – and thus enhance the signal –
we find averaging windows of 10-15 years (and sometimes longer at the smaller spatial scales)
combined with modest (nearest-neighbor) spatial averaging substantially improve the capability
for trend detection.
We show that the largest ozone variability is found in the Eastern US (Figure 5, Figure S4),
and subsequently there are many regions within the Eastern US where even a 20-year averaging





window has a non-negligible likelihood of estimating ozone variability that is dependent (with
possible error in the 1 – 3 ppbv range) on the particular years selected. In addition, over much of
the Eastern US, simulations of 5-years or shorter have a substantial likelihood (40 – 90%,
Figures S1 and S2) of reflecting the influence of meteorological variability on chemistry rather
than the mean state of surface ozone, with the possibility of 5 – 10 ppbv error (Figure S4).

Finally, we demonstrate a conceptual framework that allows for a "sliding-scale" view of

surface ozone variability, in which both temporal and spatial averaging is examined at every grid
cell within the Continental US. We show that the magnitude of estimates of ozone variability can
be reduced with both temporal and spatial averaging, although temporal averaging tends to be
more effective. While there are many regions in which both temporal and spatial averaging used
in conjunction substantially reduce the estimate of ozone variability, there are some regions
where spatial averaging is ineffective, or even counter-effective. In contrast, this is not the case
for temporal averaging, which consistently reduces the magnitude of estimated ozone variability.
Our analysis could be combined with other studies (e.g. Sofen et al., 2016) to guide
observational and modeling strategies and identify regions and scales at which particular signals
are most likely to be identified.



**Code Availability**
CESM CAM-Chem code is available through the National Center for Atmospheric Research /
University Corporation for Atmospheric Research (NCAR/UCAR) website
(http://www.cesm.ucar.edu/models/cesm1.2/), and this project made no code modifications from
the released model version.



**Data Availability**

The raw model output is archived on the NCAR servers, and processed data will be made

available upon publication on Massachusetts Institute of Technology servers.



**Supplemental Link**



**Author Contribution**
BBS ran the present-day simulation, analyzed the data, and wrote the manuscript. FGM ran the
future simulations and made the data available to BBS. NS, RP, EM, ST, and LE guided and
reviewed the scientific modeling and analysis process, and provided feedback throughout the
project and development of the manuscript.



**559**    **Competing Interests**

The authors declare that they have no conflict of interest.





**Acknowledgements**
This model development work was supported by the U.S. Department of Energy (DOE) Grant
DE-FG02-94ER61937 to the MIT Joint Program on the Science and Policy of Global Change.
Computational resources for this project were provided by DOE and a consortium of other
government, industry, and foundation sponsors of the Joint Program. For a complete list of
sponsors, see: http://globalchange.mit.edu. Additional computing resources were provided by the
Climate Simulation Laboratory at NCAR's Computational and Information Systems Laboratory
(CISL), sponsored by the National Science Foundation and other agencies.  The National Center
for Atmospheric Research is funded by the National Science Foundation. The authors would also
like to thank Daniel Rothenberg for efficient processing of the ozone files.





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



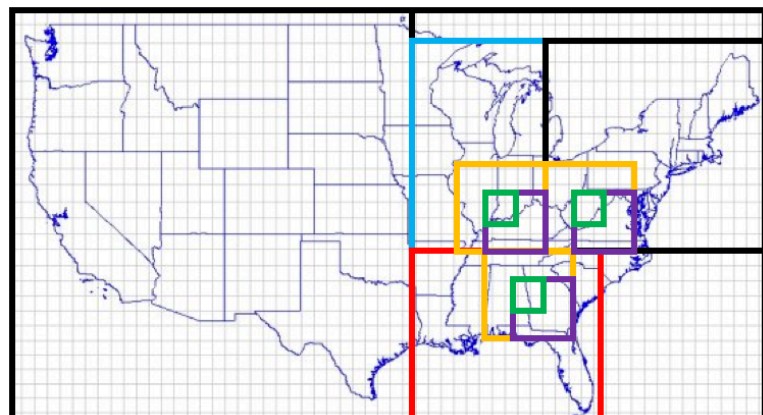

**Figure 1: Telescoping Spatial Regions included in this study. The largest scale we consider is the Continental**
**US (outer border). We focus on the Eastern US, by subdividing into three subregions: the Midwest (blue),**
**Northeast (black), and Southeast (red). Within each subregion we telescope into a 3x3 grid cell (yellow), 2x2**
**grid cell (purple), and a 1x1 grid cell (green). In the paper, we only show a subset of these telescoping regions,**
**and we include the rest in the Supplemental Material.**





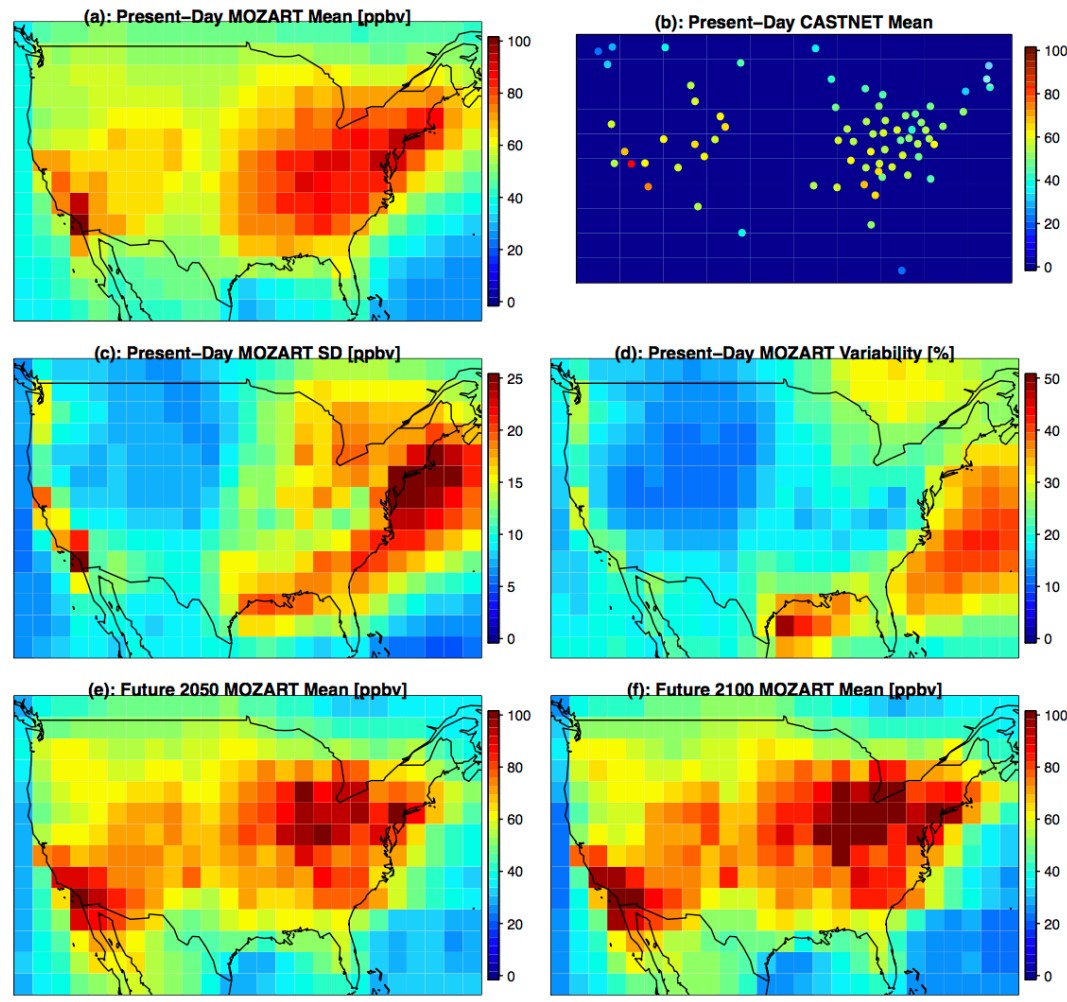

**Figure 2: Continental US surface maps of (a) present-day MOZART mean DM8H O₃; (b) present-day**
**CASTNET mean DM8H O₃; (c) present-day MOZART standard deviation; (d) present-day MOZART**
**variability (standard deviation divided by mean, as a percent); (e) future MOZART year 2050 mean DM8H**
**O₃; and (f) future MOZART year-2100 mean DM8H O₃. All model results are averaged over every JJA day**
**in the time series, while the CASTNET results are only for the year 2000.**





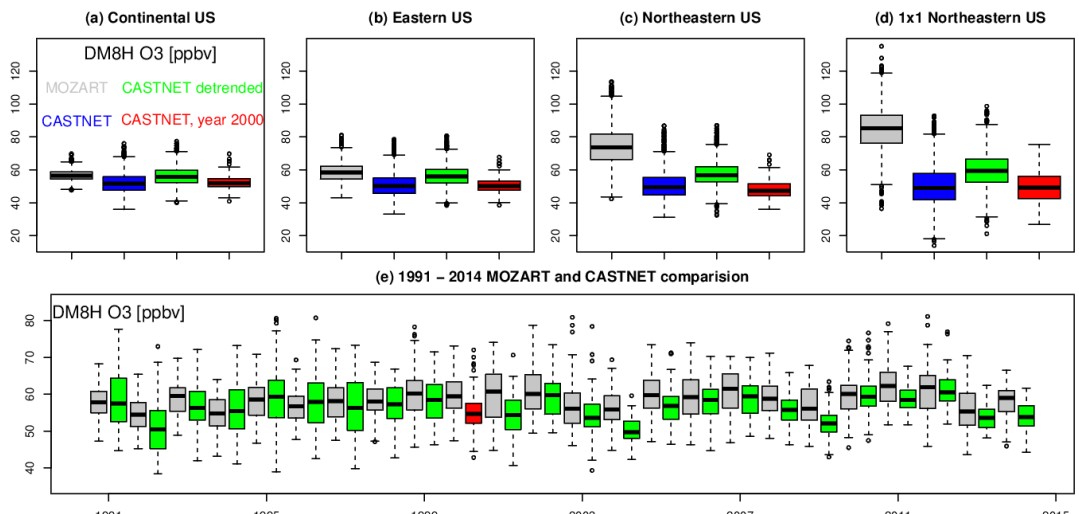

**Figure 3: (a-d): Boxplots for surface DM8H O$_3$ for every summertime (JJA) day from 1991 – 2014 averaged over the Continental US, the Eastern US, the Northeastern US, and a single grid cell in the Northeastern US from CESM CAM-chem (grey), CASTNET observations (blue), detrended CASTNET observations (green), and the detrended CASTNET values for the year 2000 only (red). (e) Comparison of the yearly JJA DM8H O3 estimates averaged over the Eastern US for MOZART (grey) and the detrended CASTNET (green) from 1991 – 2014. The single red boxplot coincides with the red boxplot in (b). The units are in ppbv, and for each boxplot the box contains the Inter Quartile Range (IQR), the horizontal line within the box is the median, and the whiskers extend out to the farthest point which is within 1.5 times the IQR with circles indicating any outliers.**





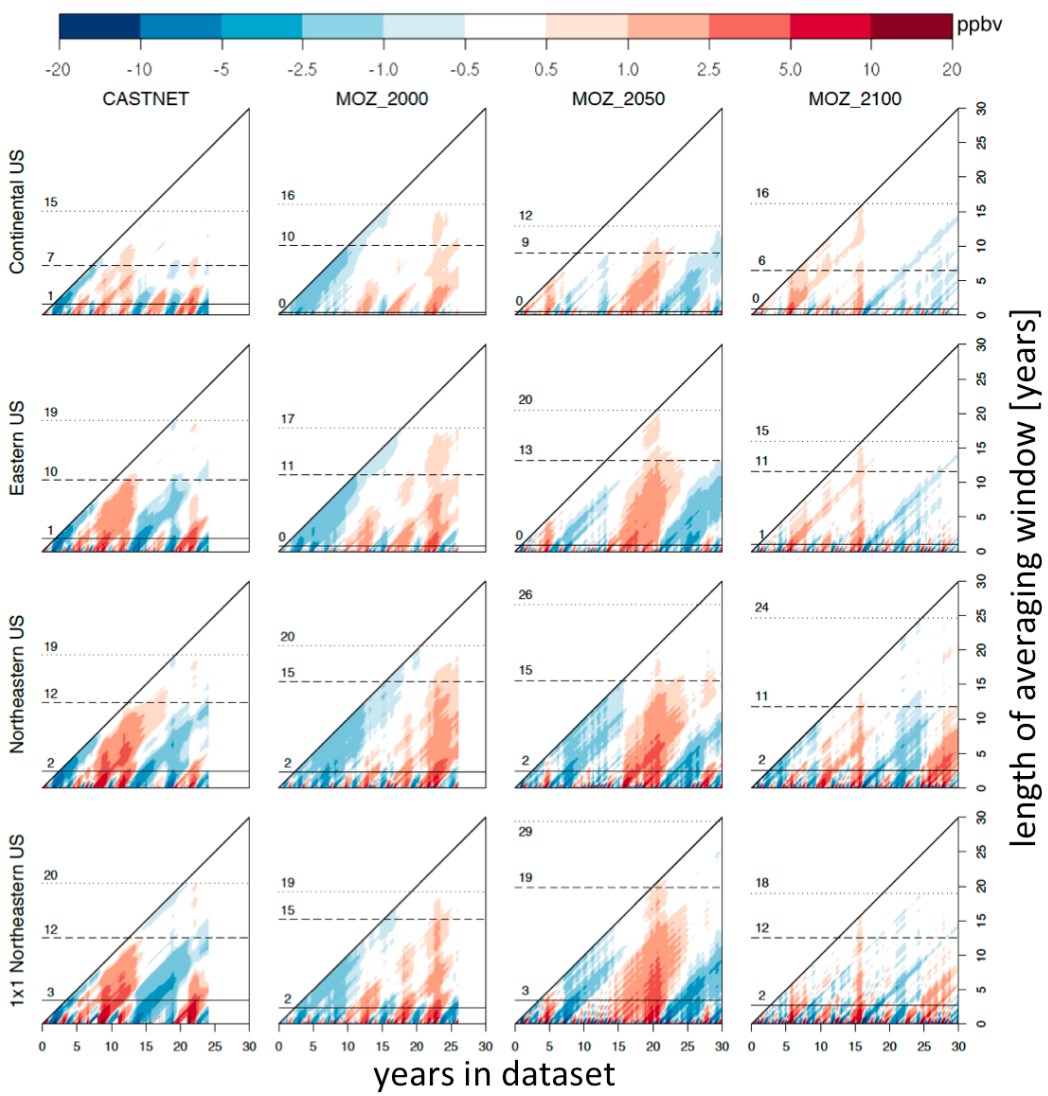

**Figure 4: A representation of the variability of the DM8H O$_3$ anomaly (from the long-term mean) for**
**the four datasets in this study (CASTNET, MOZ_2000, MOZ_2050, MOZ_2100, columns) averaged**
**over the four telescoping regions (CUS, EUS, NEUS, NEUS 1x1, rows). In each panel, the horizontal axis**
**is the number of years in the dataset (24 years (1991-2014) for CASTNET, 26 years (1990-2015) for**
**MOZ_2000, and 30 years (2036-2065 and 2086-2115) for MOZ_2050 and MOZ_2100), and the vertical**
**axis represents the length of the averaging window (ranging from 1 day (bottom row) up to the entire**
**time series (top pixel)). Each pixel represents the estimate of the ozone anomaly for a given averaging**
**window (vertical axis) ending at a given time (horizontal axis). Horizontal lines indicate the length of**
**averaging window required to guarantee that the variability drops below thresholds of 5 ppbv (solid), 1**
**ppbv (dashed), and 0.5 ppbv (dotted).**



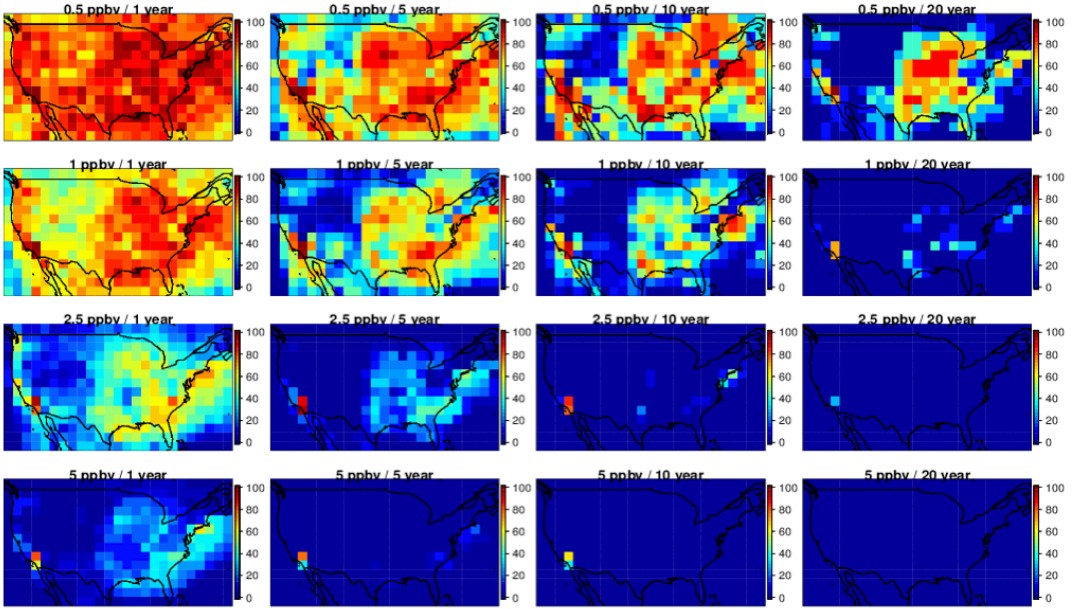

**Figure 5: Spatial Plots over the Continental US plotting the likelihood (%) that an estimate of ozone**
**exceeds a given threshold due to meteorological variability (rows) at the grid-cell level when using**
**different lengths of averaging windows (columns) for the present-day CESM simulation (MOZ_2000).**

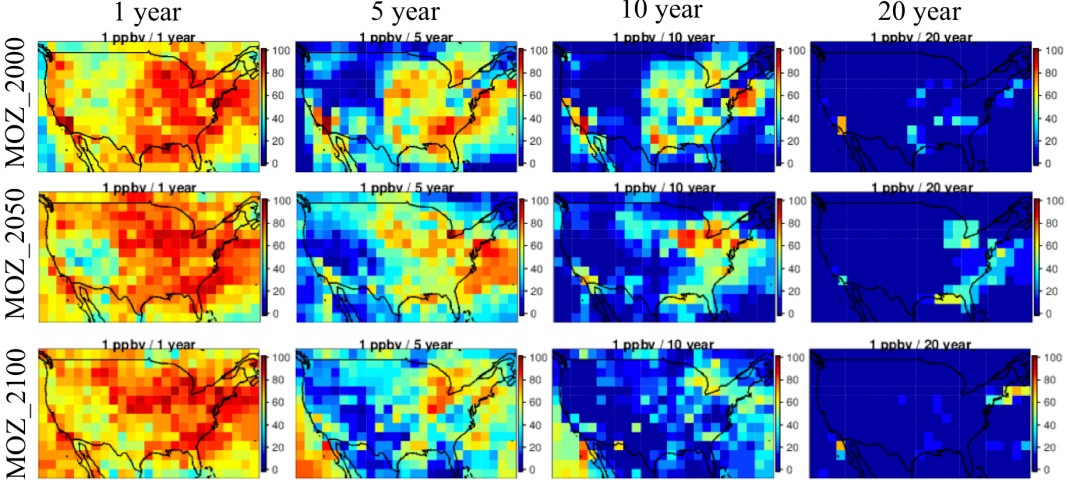

**Figure 6: As in Figure 5, but only the second row, for present-day CAM-chem, future CAM-chem 2050,**
**and future CAM-chem 2100.**



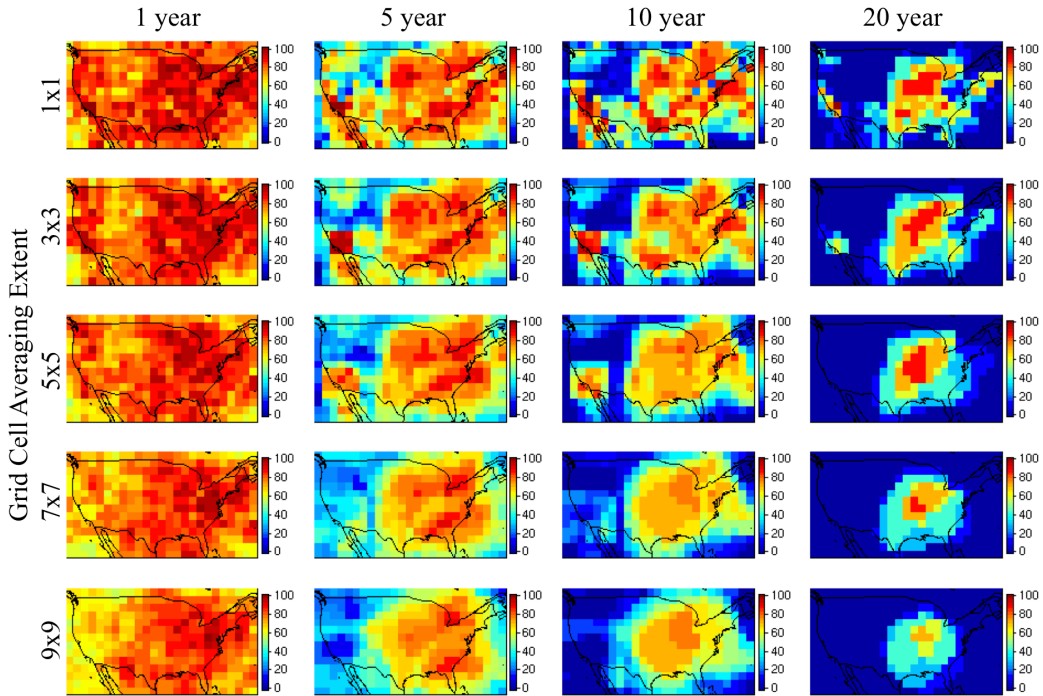

**Figure 7: Combined impact of temporal and spatial averaging on reducing ozone variability on the likelihood (%) of exceeding the 0.5 ppbv threshold (as in Figures 5, 6, and Supplemental Figure S3) for the present-day MOZ_2000 simulation. The top row is the same as in Figure 6, while the lower rows have averaged the values within a 3x3, 5x5, 7x7, and 9x9 box surrounding each individual grid cell.**
806





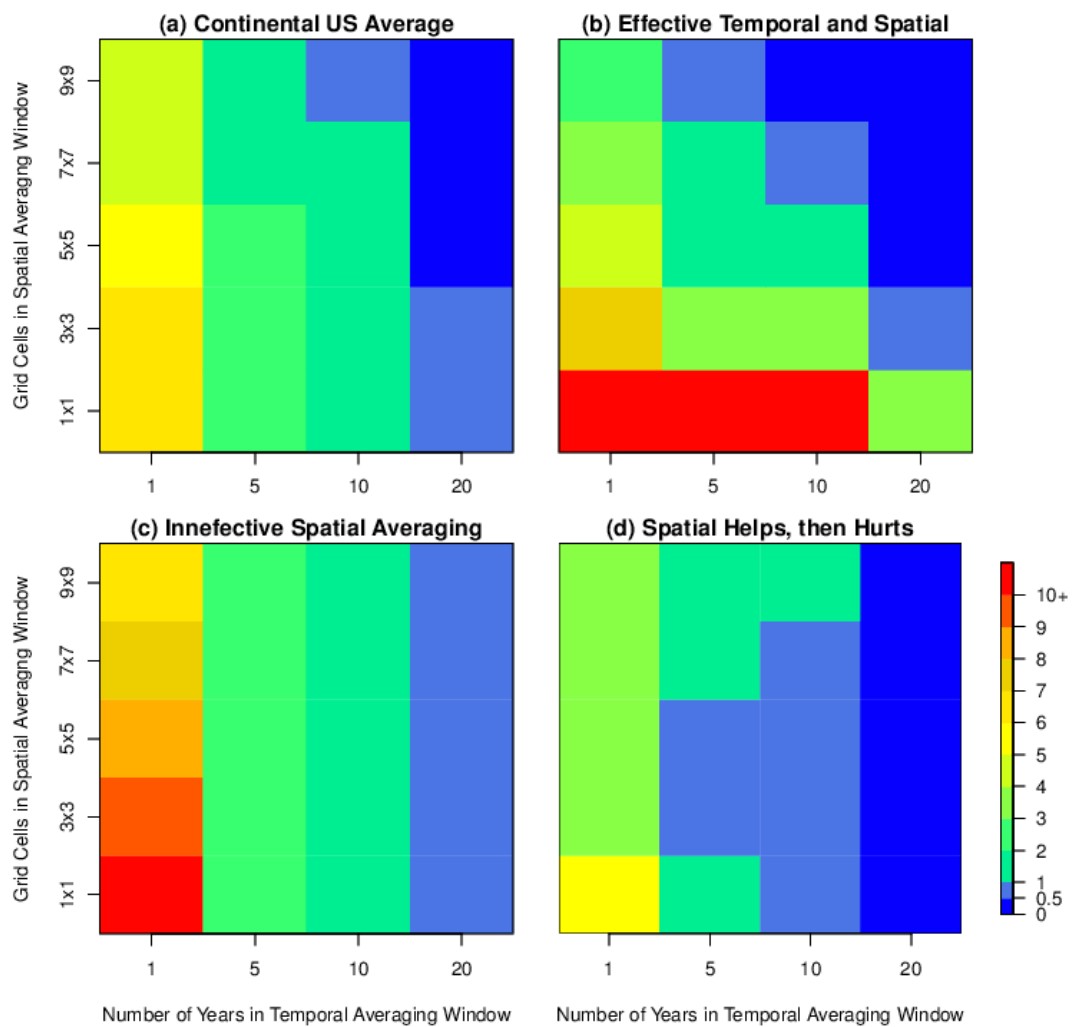

807

**Figure 8: The maximum potential calculated DM8H O$_3$ anomaly [ppbv] from the long-term mean for (a) the Continental US average and three individual grid cells taken from (b) Southern California, (c) the Northeast, and (d) the Rocky Mountains demonstrating the impact of temporal and spatial averaging, with the number of years included in the temporal averaging window increasing along the x-axis and the number of grid cells included in the spatial averaging window increasing along the y-axis. A full map of the Continental US can be found in the Supplemental Material (Figure S4). Note that the color scale is non-linear, and the color transitions are selected to match the thresholds established throughout this paper.**




|  |  |  | CASTNET | MOZ_2000 | MOZ_2050 | MOZ_2100 |
|---|---|---|---|---|---|---|
| **Continental US** | **Mean** | **ppbv** | 52.4 | 56.7 | 56.8 | 57.4 |
|  | **Standard Deviation** | **ppbv** | 5.04 | 3.08 | 3.54 | 3.73 |
|  | **Variability** | **%** | 10% | 5% | 6% | 7% |
|  | **Bias** | **ppbv** |  | 4.31 |  |  |
| **Eastern US** | **Mean** | **ppbv** | 50.7 | 58.6 | 55.5 | 56.5 |
|  | **Standard Deviation** | **ppbv** | 5.78 | 5.77 | 5.80 | 6.50 |
|  | **Variability** | **%** | 11% | 10% | 10% | 12% |
|  | **Bias** | **ppbv** |  | 7.91 |  |  |
| **Northeastern US** | **Mean** | **ppbv** | 48.3 | 74.4 | 68.4 | 73.0 |
|  | **Standard Deviation** | **ppbv** | 6.89 | 11.4 | 11.1 | 12.7 |
|  | **Variability** | **%** | 14% | 15% | 16% | 17% |
|  | **Bias** | **ppbv** |  | 26.1 |  |  |
| **1x1 Northestern US** | **Mean** | **ppbv** | 49.6 | 84.9 | 81.1 | 85.1 |
|  | **Standard Deviation** | **ppbv** | 10.2 | 12.8 | 16.7 | 17.3 |
|  | **Variability** | **%** | 21% | 15% | 21% | 20% |
|  | **Bias** | **ppbv** |  | 35.3 |  |  |

**Table 1: Statistical Summary of the CASTNET observations and the three CAM-chem simulations for**
**different spatial averaging regions within the US. Variability is defined as the standard deviation**
**divided by the mean value (in percent). Biases are only included for the present-day CAM-chem**
**simulation compared to the CASTNET data. Similar tables for the other regions in this study are**
**included in the Supplemental Material.**