# Peer review of "Maximizing Ozone Signals Among Chemical, Meteorological, and Climatological Variability"

_Atmospheric Chemistry and Physics, 2017_

## Referee Comment (RC1) · Anonymous Referee #1 · 13 Dec 2017

General comments:

This paper discusses the use of different temporal and spatial averaging scales to detect trends in surface ozone over the United States. This is an interesting topic that is useful to the community, and the approach is novel. However, I have two general concerns that I would like to see addressed:

1. The relevance of the particular methods discussed for detection of air quality trends should be better clarified or caveated, since the averaging time-scales suggested (10-15 years) are comparable to the trends we seek to detect, and temporal and spatial averaging can blur localized signals of high ozone that are relevant to public health.

[Figure]

2. Given the heavy dependence of the analysis on model simulations, I would like to see more rigorous evaluation of the model's ability to accurately predict the spatial and temporal variability of surface ozone and its response to changes in meteorology and emissions.

In addition, a number of other statistical techniques have been applied to the problem of separating emission effects from other drivers of variability (for example, Camalier et al., Atmos. Environ., 2007, and refs thereien), with the potential advantage of detecting changes on shorter timescales. How do the results in this paper compare to other statistical methods? Perhaps this could be discussed in the discussion or conclusion sections.

Specific comments:

Line 28: How is the "chemical variability" that is not related to "meteorological variability" different from an air quality signal?

Line 41: The authors state on line 31 that part of the motivation for this study is to identify the impact of emission reduction policies on e.g. ozone. Here, however, they suggest averaging over 10-15 years. This seems pretty long compared to the timescale of air quality changes and compared to the available data records, which for many CASTNET sites in only on the order of 20 years.

Line 42: If you average over several hundred kilometers, do you risk missing policy or health-relevant ozone exceedences that occur at more local scales?

Line 66: For signal detection, see also Weatherhead et al., Physics & Chemistry of Earth, 2002; Strode and Pawson, JGR, 2013; Deser et al., Climate Dynamics, 2011

Lines 96-110: While it is true that the 4th highest MDA8 criteria includes some averaging, it is also aimed at capturing the high end of the distribution rather than just the long-term mean. Isn't this lost by simply averaging over longer periods?

Line 148: Since you are interested in different spatial scales, why not include urban air

quality sites as well as CASTNET?

Line 205: Please highlight the key differences between this and the earlier model version.

Line 252: The assertion that the spatial variability is well-captured is not really evident in Figure 2. Maybe overplot the observations on top of the model map, or report the spatial correlation between the model and the observations.

Section 3.1 and Fig. 2: It would be helpful to show the temporal variability of the observations along side that of the model

Line 255: Clarify that it is the standard deviation in the model.

Line 272: What is the correlation between the modeled and observed timeseries? Figure 3e suggests a lot of mismatches between the observations and model. What does this mean in terms of the uncertainty in your model-based findings?

Section 3.2, first paragraph: Some of this could go in the methods section.

Lines 374-375: Can you explain why? Do these regions have higher variability?

Line 430: The relationship between chemical and meteorological variability also depends on emission levels (e.g. Bloomer et al., GRL, 2009), and these are unlikely to remain constant over a decadal averaging window. Thus the real situation will be more complicated than the constant-emission model-based analysis shown here. The model-based analysis is still useful, but should be more carefully caveated.

Technical:

Line 374: "Shorter" not "short"

---

## Referee Comment (RC2) · Anonymous Referee #2 · 15 Feb 2018

Review of "Maximizing ozone signals among chemical, meteorological, and climatological variability" by B. Brown-Steiner et al. (acp-2017-954)

General Comments ——————-

This manuscript describes an evaluation of the variability of surface ozone concentrations over the United States during summer. In particular, the authors analyze the effects of meteorological variability on ozone concentrations, and the dependence of this variability on temporal and spatial averaging scales. The goals is to use averaging to provide a more robust estimate of the uncertainty in the "true" ozone concentration, independent of the influence of meteorological "noise". The idea that spatial or tem-

poral averaging can reduce meteorological variability is not a new one, but this paper presents a useful and innovative framework for analyzing the choice of time and space scales, depending on the uncertainty threshold required for a particular application. This writing in this paper could be improved significantly to clarify the methods used and the basis for the recommendations being made. I list below some such suggestions for ways the manuscript can be improved. With revisions, this paper would be appropriate for publication in ACP, and would be a helpful contribution to the literature on detecting robust signals in ozone over a noisy background.

Specific Comments ——————

Abstract

line 41 – This 10-15 year time period pertains to detecting a robust estimate of mean ozone concentrations. What are the implications for detecting trends (e.g., driven by emission changes) in ozone? For instance, large robust trends in ozone were detected in observations as a result of emission reductions following the NOX SIP Call. This manuscript claims to provide information on estimating trends in ozone, but does not really provide specific information on trend detection methodologies.

lines 44-46 – For which other quantities might these results be applicable? What features of the spatiotemporal distribution dictate the choice of optimal spatial and temporal averaging periods.

1. Introduction

lines 93-95 – Mention also internal (unforced) variability.

lines 91-97 – There is not a clean distinction between running ensembles of model runs with different initial conditions versus "expand[ing] the temporal averaging window". In the case of "climatological" runs such as those done here with CAM-Chem, running more years in a single simulation is nearly identical in practice to running more years of a single simulation.

lines 123-125 – You mention here that the objective is to "limit the likelihood of over-confidence in an estimate of surface ozone". Presumably, the goal is more than that. Rather than just providing an improved (large) estimate of local variability, the averaging method suggested here also aims to reduce the underlying uncertainty due to meteorological variability.

lines 154-155 – Model resolution is not addressed in this study. How would varying model resolution compare with the other "parametric" changes in the model discussed here?

2.1 CAM-Chem

In this section and throughout the paper, the model name "MOZART" seems to be used interchangably with "CAM-chem", including in the names of the simulations. This is confusing, since MOZART and CAM-chem, although closely related, are distinct models. Please clarify throughout the paper.

line 200 – Here and elsewhere throughout the paper, clarify that you are only considering the effect of future *climate*, not actually fully simulating future conditions (e.g., future emissions).

2.3 Telescoping Regional Definitions

lines 230-232 – This sentence is repetitive of Intro.

3.1 Spatial and Temporal Comparisons

line 248 – Throughout the paper, the notation "DM8H" is used for the daily maximum 8-hour ozone concentration. Elsewhere in the literature, this seems to be referred to as "MDA8".

line 248 – "MOZART" –> "CAM-chem"

lines 255-259, Figure 2 – Show standard deviation and/or variability from the observations as well. If the standard deviation were similar between the model and observa-

tions, would the model ozone bias cause the (relative) variability to differ significantly?

line 283 – Add "(Figure 2, Table 1)" after "Here".

lines 283-285 – This sentence is repetitive of the first paragraph in this section.

line 289 – Add "from continental to a single NE U.S. grid box" after "telescoping regions".

line 290 – Add "albeit with lower overall variability" after "captures this trend".

3.2 Variability, Averaging Windows, and Thresholds

line 314 – Add "underlying variability at the" before "particular choice of spatial and temporal scale".

line 328 – Does "variability" here refer to standard deviation (as suggested by the ppbv thresholds) or as previously used, the relative variability (s.d./mean)? Confusing. Make sure to define the quantities being discussed.

line 329 – Clarify what is meant here by "This difference".

3.3 Selection of Temporal Averaging Scales

line 358-359 – Add "meteorological variability causing ozone anomalies" before "exceeding particular thresholds", if this is the intended meaning.

line 363 – "Increas[ing] the threshold" is not really a strategy for "filtering out the noise". It is more like accepting the higher level of noise.

lines 367 -370 – Confusing as written. Separate out the mention of Fig.S3 to a second sentence, e.g., "Similarly, in Supplemental Figure S3, one column (the 5-year averaging window) is selected."

line 369 – "Figure 6" –> "Figure 5"

line 369 – Add "compare with" before "equivalent plots".

line 370 – "Figures 7" –> "Figures 6".

4. Discussion

line 434 – Add "variability" after "surface ozone".

line 460 – Cut comment in parentheses about future simulations. It is not known whether the future simulations will/would exhibit biases.

5. Conclusions

line 502 – Add "and" after "configurations".

line 506 – Add "summertime" before "surface ozone". Clarify throughout conclusions that the analysis presented here is restricted to summer.

line 513 – Add "summertime" before "ozone variability".

line 523 – As mentioned earlier, the discussion of trend detection in the manuscript is very weak. Much more could (and should) be said about the application of the averaging methods presented here for trend detection. For instance, what are the implications of needing 10-15 year averaging windows for the length of timeseries needed to detect ozone trends (e.g., forced by climate change or emissions changes)?

lines 524-530 – Mention here the compounding of (meteorological) variability in the observations with changes caused by variability/trends in emissions.

Figure 2 – Add the standard deviations plotted here standard deviations of daily ozone concentrations? If so, then for comparison with Figure 5, it would be useful also to show the interannual standard deviation of seasonal mean ozone.

Figure 3 – Explain that the CAM-chem simulation has fixed year-2000 emissions and SST, but time-varying meteorology. Why are the CASTNET values for 2000 "detrended", instead of showing raw 2000 values? Change "MOZART" to "CAM-chem". In legend text in panel (a), also change "MOZART" to "CAM-chem".

Figure 4 – Define what is meant here by "variability". Is it the standard deviation, or the relative variability (s.d./mean)? Mention in caption that this plot shows summer ozone only. This is confusing from how the vertical axis is plotted.

Figure 8 – Change panel titles to the names of the regions. Keep the description of the regimes for filtering effectiveness in the text instead.

---

## Author Comment (AC1) · 29 Mar 2018

Response to Reviewers for manuscript 'Maximizing Ozone Signals Among Chemical, Meteorological, and Climatological Variability' (https://www.atmos-chem-phys-discuss.net/acp-2017-954/)

We would like to thank the reviewers for their valuable comments.

Below we work through each of the reviewers' comments, with the comments in black and our responses in red. We also include any alterations to the text in red after our responses with the specific additions indicated with underlines. Line references refer to the tracked changes document.

**Anonymous Referee #1

**General comments:**

This paper discusses the use of different temporal and spatial averaging scales to detect trends in surface ozone over the United States. This is an interesting topic that is useful to the community, and the approach is novel. However, I have two general concerns that I would like to see addressed:

1. The relevance of the particular methods discussed for detection of air quality trends should be better clarified or caveated, since the averaging time-scales suggested (10- 15 years) are comparable to the trends we seek to detect, and temporal and spatial averaging can blur localized signals of high ozone that are relevant to public health.

This is a very valid point, and one that we underemphasized in the original manuscript. We have updated portions of this manuscript to focus more on 'signals' rather than 'trends' (as we include trends as one type of signals). We add language to the discussion (Line 521-524 and 591-594) and conclusions (Lines 611-651, 615-616, 633-644, 652-656) that highlights the difficulty in balancing data availability, observation/simulation length, averaging times, and error thresholds. See our responses to the specific comments below for details on these additions.

2. Given the heavy dependence of the analysis on model simulations, I would like to see more rigorous evaluation of the model's ability to accurately predict the spatial and temporal variability of surface ozone and its response to changes in meteorology and emissions.

The CESM1.2 CAM-chem model has been extensively evaluated in previous papers mentioned in the methods section. We have added more explicit references to this evaluation throughout the manuscript (Lines 201-202 and 221-222). We have added additional evaluation of the model capabilities compared to the available observations with regard to meteorological variability (updated Figure 2 and reference to Brown-Steiner et al, in review, see following paragraph). We do not examine the impact of emissions variability in this manuscript, as this is beyond the scope of the current work, but we add additional emphasis in the conclusions that emissions variability studies are needed in future research:

Lines 618-620: *"Taking into account the complex interactions involving trends and variability between emissions, chemistry, meteorology, and climatology necessitates a variety of strategies."*

Lines 652 – 656: *"While we have detrended the CASTNET observations to compare to the constant year-2000 cycled emissions in the simulations, the CASTNET time series inherently includes the compounded variability of both meteorological and emission sources. Future studies will need to expand this analysis to include trends and variability in the emissions, as well as in the meteorology."*

In addition, these model runs (along with others) are more thoroughly compared to observations in a second paper which is now in discussion in GMD, and we have added a reference to this paper to this manuscript: Brown-Steiner, B., Selin, N. E., Prinn, R., Tilmes, S., Emmons, L., Lamarque, J.-F., and Cameron-Smith, P.: Evaluating Simplified Chemical Mechanisms within CESM Version 1.2 CAM-chem (CAM4): MOZART-4 vs. Reduced Hydrocarbon vs. Super-Fast Chemistry, Geosci. Model Dev. Discuss., https://doi.org/10.5194/gmd-2018-16, in review, 2018.

In addition, a number of other statistical techniques have been applied to the problem of separating emission effects from other drivers of variability (for example, Camalier et al., Atmos. Environ., 2007, and references therein), with the potential advantage of detecting changes on shorter timescales. How do the results in this paper compare to other statistical methods? Perhaps this could be discussed in the discussion or conclusion sections.

We have added language in the conclusion that contrasts our methodologies to other methodologies, and encourages a multi-strategy approach:

Lines 615-622: "Our analysis and conceptual framework *presented here cannot solve this tension, but it does demonstrate some strategies which can* allow for a selection of spatial and temporal averaging scales*, and a consideration of the error threshold,* that can aid in this signal detection *on a case-by-case basis. Taking into account the complex interactions involving trends and variability between emissions, chemistry, meteorology, and climatology necessitates a variety of strategies. This work quantifies the impact of spatial and temporal averaging in signal detection, which can be used in conjunction with ensembles of simulations, statistical techniques, and other strategies to further out understanding of the chemical variability in our atmosphere.*"

We have also included Camalier et al., 2007 (Line 71) and other recommended citations from below (Line 69, Line 71) to the introduction in order to provide as much information to readers about the possible strategies and methodologies used for signal detection.

**Specific comments:**

Line 28: How is the "chemical variability" that is not related to "meteorological variability" different from an air quality signal?

Chemical variability can result from more than just meteorological variability (e.g. emissions variability, which we do not address in this paper) and also non-linear interactions between chemistry, emissions, meteorology, climatology, and surface processes. We have clarified this in the abstract:

Line 28-30: "However, the magnitude of a surface air quality signal is generally small compared to the magnitude of the underlying *chemical, meteorological, and climatological variabilities (and their interactions) that exist both in space and in time, and which include variability in emissions and surface processes.*"

Line 41: The authors state on line 31 that part of the motivation for this study is to identify the impact of emission reduction policies on e.g. ozone. Here, however, they suggest averaging over 10-15 years. This seems pretty long compared to the timescale of air quality changes and compared to the available data records, which for many CASTNET sites in only on the order of 20 years.

We recognize that our suggestion of averaging over 10-15 years is challenging, but it is consistent with recent literature (e.g. Barnes et al. 2015; Garcia-Menendez et al. 2017). We hope with this manuscript to demonstrate some of the difficulties that arise when trying to detect the impact of, for instance, emissions reduction policies on ozone. In particular, we hope to demonstrate that the variability in atmospheric chemistry needs to be quantified, examined, and addressed in a direct manner when identifying signals, and that the temporal and spatial context of the particular signal needs to be provided as supporting evidence that a particular signal is robust. We have added the following sentences to the conclusion section to emphasize this point.

Lines 633-641: *"We recognize that achieving a 10 – 15 year temporal averaging window is difficult, but this recommendation is consistent with recent literature (e.g. Barnes et al., 2015; Garcia-Menendez et al., 2017). For studies where 10 – 15 years of averaging is impractical, we recommend that some spatial and temporal context is provided that demonstrates that the signals being examined are robust and not the result of internal variability or noise."*

We also add language on Lines 43-46 and 611-615, in response to additional reviewer comments below.

Line 42: If you average over several hundred kilometers, do you risk missing policy or health-relevant ozone exceedences that occur at more local scales?

Absolutely! Again, we hope to demonstrate the challenges of identifying chemistry signals at small spatial scales. In particular, if you are examining signals and the smallest spatial scales, it is likely a longer temporal period will be required to 'escape' the variability at that scale. We address this in the Discussion Section, in particular Line 513 in asking "What is the magnitude of ozone variability due to meteorology alone at the smallest spatial scale?" To further clarify this in the manuscript, we added the following to the abstract:

Lines 43-46: *"If this level of averaging is not practical (e.g. the signal being examined is*

*at a local scale), we recommend some exploration of the spatial and temporal variability to provide context and confidence in the robustness of the result."*

Line 66: For signal detection, see also Weatherhead et al., Physics & Chemistry of Earth, 2002; Strode and Pawson, JGR, 2013; Deser et al., Climate Dynamics, 2011

These citations have been added as further examples of the history and difficulties in signal detection into the introduction, as Camalier et al., 2007 (Lines 69 and 71).

Lines 96-110: While it is true that the 4th highest MDA8 criteria includes some averaging, it is also aimed at capturing the high end of the distribution rather than just the long-term mean. Isn't this lost by simply averaging over longer periods?

It is, and the 4$^{th}$ highest MDA8 metric has been designed for a practical legal purpose. It is also a standard metric used throughout the literature, so we felt that examining the impact of spatial and temporal averaging on this metric would be an appropriate way of adding to the literature, and the broader context of this particular metric. We add the following sentence to address this:

Lines 286-289: "*Some of the averaging strategies we present can average away the high ozone behavior this MDA8 O$_3$ metric is intended to quantify, but it is such a well-reported metric that focusing our analysis on it allows for ready comparisons to other studies.*"

Line 148: Since you are interested in different spatial scales, why not include urban air quality sites as well as CASTNET?

Since the CASNTET observations are from more rural sources, they are generally accepted as more appropriate to compare to coarse-grid cell models such as CAM-Chem. We add additional references to other studies that used CASTNET observations in this way:

Line 240: "*(e.g. Brown-Steiner et al., 2015; Phalitnonkiat et al., 2016)*"

Future research should extend analysis like that presented here to models of different resolutions (and the associated observations). We have added this suggestion to the last paragraph of the Discussion Section:

Lines 591-594: "*Furthermore, future research examining the impact of spatial and temporal averaging using regional-scale models, models with different resolutions, and the inclusion of urban observations could provide additional insight into understanding chemical variability and averaging techniques.*"

Line 205: Please highlight the key differences between this and the earlier model version.

The Tilmes et al. (2015) reference (and references therein) fully documents CESM1.2, although we had previously omitted the reference from this sentence. It has been added in Lines 191, 198, and 221.

Line 252: The assertion that the spatial variability is well-captured is not really evident in Figure 2. Maybe overplot the observations on top of the model map, or report the spatial correlation between the model and the observations.

We have updated Figure 2b to better compare model/observations. We also add a reference in the caption to Brown-Steiner et al. (in review, GMDD), which performs additional model-observation comparisons of CAM-Chem.

Section 3.1 and Fig. 2: It would be helpful to show the temporal variability of the observations along side that of the model

Since we only compare the year 2000 in this figure (this was not clear in the caption and this has been updated), there is not enough data for a full comparison of temporal variability, but Figure 2b has been updated and additional references to Brown-Steiner et al. (in review, GMDD) on Line 898, which extends model-observation comparisons for MOZART-4 (and other mechanisms).

Line 255: Clarify that it is the standard deviation in the model.

We have now clarified this as follows:

Line 301: "…The standard deviation *of the simulated MDA8 $O_3$*…"

Line 272: What is the correlation between the modeled and observed timeseries? Figure 3e suggests a lot of mismatches between the observations and model. What does this mean in terms of the uncertainty in your model-based findings?

Model correlations to observations depend on the region (with $R^2$ values ranging from 0.42 – 0.88, see the updated Figure 2b), and Figure 3e compares the average over the entire Eastern US. Comparisons of the seasonal correlations to observations are available in Brown-Steiner et al. (in review, GMDD) and are generally high for MOZART CAM-chem (0.8 – 0.9). References to this paper are added to this manuscript (Lines 198, 201-202, Figure 2 Line 898). Since we include cycled year 2000 emissions for out simulations, we do not expect a high correlation for the entire time series, even when compared to the detrended CASTNET observations since we do not simulate the real-world emissions variability, especially when comparing individual sites to model grid boxes. This additional uncertainty that comes from assuming cycled emissions has been noted in other comments, and additional language has been put in the Discussion and Conclusions to explore the implications (Lines 521-524 (see below), 611-615 (see above), 652-656 (see below)).

Section 3.2, first paragraph: Some of this could go in the methods section.

We moved up the more technical description to the newly added Section 2.4 (Line 257-274).

Lines 374-375: Can you explain why? Do these regions have higher variability?

Yes, this has been clarified and the reader is pointed to Figure 2d:

Lines 446-448: "*Shorter* windows (or smaller thresholds) are needed in the Western US *(where variability is smaller, see Figure 2d)* than in the Eastern *US (where variability is larger) as well* as over coastal and highly populated regions."

Line 430: The relationship between chemical and meteorological variability also depends on emission levels (e.g. Bloomer et al., GRL, 2009), and these are unlikely to remain constant over a decadal averaging window. Thus the real situation will be more complicated than the constant-emission model-based analysis shown here. The model-based analysis is still useful, but should be more carefully caveated.

We add additional text at the end of this paragraph to caveat the limits of our methodology and highlight the complexities that arise when considering trends and variability in emissions, meteorology, and climate:

Lins 521-524: "*A more comprehensive analysis of chemical variability will need to account for both meteorological and emission variability, which is complicated by temporal trends in both the emissions of ozone precursor species and the climate.*"

Technical: Line 374: "Shorter" not "short"

This has been corrected Line 446).

**Anonymous Referee #2

**General Comments ——————-**

This manuscript describes an evaluation of the variability of surface ozone concentrations over the United States during summer. In particular, the authors analyze the effects of meteorological variability on ozone concentrations, and the dependence of this variability on temporal and spatial averaging scales. The goals is to use averaging to provide a more robust estimate of the uncertainty in the "true" ozone concentration, independent of the influence of meteorological "noise". The idea that spatial or temporal averaging can reduce meteorological variability is not a new one, but this paper presents a useful and innovative framework for analyzing the choice of time and space scales, depending on the uncertainty threshold required for a particular application. This writing in this paper could be improved significantly to clarify the methods used and the basis for the recommendations being made. I list below some such suggestions for ways the manuscript can be improved. With revisions, this paper would be appropriate for publication in ACP, and would be a helpful contribution to the literature on detecting robust signals in ozone over a noisy background.

**Specific Comments ——————**

Abstract

line 41 – This 10-15 year time period pertains to detecting a robust estimate of mean ozone concentrations. What are the implications for detecting trends (e.g., driven by emission changes) in ozone? For instance, large robust trends in ozone were detected in observations as a result of emission reductions following the NOX SIP Call. This manuscript claims to provide information on estimating trends in ozone, but does not really provide specific information on trend detection methodologies.

We explore some of the literature on ozone trends in the introduction (Cooper et al., 2012, Barnes et al., 2016, and others), and although we do not provide specific trend detection methodologies, we feel that we have demonstrated the potential risks of calculating trends based on an individual selection of years. You are correct in that we use the word 'trend' in many places where we really mean 'signal,' so we have changed the word 'trend' to 'signal' in several of these places throughout the manuscript to better reflect our intended message: the description of signals that we present in the introduction.

We have also added language (addressing other comments) that address the implications of the 10 – 15 year time period throughout the manuscript (Lines 46 and 633-644, addressed in previous comments, and Lines 611-615):

Lines 611-615: "*In particular, it would be impractical to delay interpreting observations for 10 – 15 years, or alternatively to expand the spatial averaging such that small-scale features are smoothed away. Nonetheless, it is unwise to over-interpret trends and signals based on observations from a limited spatial area and over a short temporal*

*period.*"

lines 44-46 – For which other quantities might these results be applicable? What features of the spatiotemporal distribution dictate the choice of optimal spatial and temporal averaging periods.

Those are excellent questions and we intentionally left this open to the reader. Naturally, this analysis could apply to other chemical species, but also chemistry-meteorology interactions (e.g. ozone-temperature relationship), surface features (land use cover, plant functional type, surface roughness, albedo, cloud and boundary layer variables, etc). We add the following to the discussion section, indicating some quantities that this strategy may apply to:

Lins 580-584: *"In particular, low-frequency oscillations (e.g. ENSO, and others) and other forms of internally or externally forced trends (e.g. anthropogenic and natural changes in emissions) are readily adaptable to this type of analysis, which could address signals pertaining to precipitation, biogenic emissions, boundary layer variables, cloud properties, and many others."*

1. Introduction

lines 93-95 – Mention also internal (unforced) variability.

Added:

Line 99-101: "This approach cannot address structural uncertainties *and internal (unforced) variability* between models, but is capable of identifying parametric uncertainties within a single model."

lines 91-97 – There is not a clean distinction between running ensembles of model runs with different initial conditions versus "expand[ing] the temporal averaging window". In the case of "climatological" runs such as those done here with CAM-Chem, running more years in a single simulation is nearly identical in practice to running more years of a single simulation.

We agree. There are many modeling choices (ensembles with different initial conditions and internally simulated meteorology, ensembles with internally simulated meteorology and different emissions (either transient or cycling a single year), ensembles with forced meteorology and different emissions, ensembles with different sets of online/offline forcing datasets (oceans, ice, land, etc.). What we have done in this paper is one strategy, and we hope that future studies will select other strategies. We have added the following sentence to the conclusion to indicate that what we present is one strategy among many:

Lines 641-644: *"We also recognize that our analysis is just one strategy for enhancing signal detection capabilities, and will ideally be used alongside others, such as perturbed initial condition ensembles, running simulations with either internal or forced meteorology, and examining a region or time period with different models or parameterizations."*

lines 123-125 – You mention here that the objective is to "limit the likelihood of over-confidence in an estimate of surface ozone". Presumably, the goal is more than that. Rather than just providing an improved (large) estimate of local variability, the averaging method suggested here also aims to reduce the underlying uncertainty due to meteorological variability.

Yes, this has been added:

Lines 130-132: "Our objective in this study is to provide a framework for selecting spatial and temporal averaging scales *that reduces the uncertainty in analyzing ozone signals and* limits the likelihood of over-confidence in an estimate of surface ozone that arises from meteorological variability."

lines 154-155 – Model resolution is not addressed in this study. How would varying model resolution compare with the other "parametric" changes in the model discussed here?

That is an excellent question that was outside of the scope of this paper, but we have added this as a path for future research at the end of the Discussion Section:

Line 585: "*Furthermore, future research examining the impact of spatial and temporal averaging using regional-scale models, models with different resolutions, and the inclusion of urban observations could provide additional insight into understanding chemical variability and averaging techniques.*"

2.1 CAM-Chem

In this section and throughout the paper, the model name "MOZART" seems to be used interchangably with "CAM-chem", including in the names of the simulations. This is confusing, since MOZART and CAM-chem, although closely related, are distinct models. Please clarify throughout the paper.

Throughout the manuscript, we have updated the descriptions. We leave in the name MOZART when we are specifically referencing the chemical mechanism and CAM-chem when we are more generally talking about the simulation. This has been made explicit in the methods section:

Lines 196-198: "We conduct our simulations using the MOZART-4 chemical mechanism (Emmons et al., 2010)*, which is a full tropospheric chemical mechanism integrated into CAM-Chem (e.g. Brown-Steiner et al., in review).*"

line 200 – Here and elsewhere throughout the paper, clarify that you are only considering the effect of future *climate*, not actually fully simulating future conditions (e.g., future emissions).

We have clarified this on Line 215 ("…*We also include two reference simulations of the future climate,* …") and throughout the manuscript.

2.3 Telescoping Regional Definitions lines 230-232 – This sentence is repetitive of Intro.

This sentence has been removed.

3.1 Spatial and Temporal Comparisons

line 248 – Throughout the paper, the notation "DM8H" is used for the daily maximum 8-hour ozone concentration. Elsewhere in the literature, this seems to be referred to as "MDA8".

DM8H has been changed to MDA8 throughout the manuscript.

line 248 – "MOZART" –> "CAM-chem"

We have corrected this here and throughout the manuscript.

lines 255-259, Figure 2 – Show standard deviation and/or variability from the observations as well. If the standard deviation were similar between the model and observations, would the model ozone bias cause the (relative) variability to differ significantly?

Figure 2b has been updated with a direct comparison between the model and the observations. The standard deviation comparison between the model and the observations again depends on the region. Table 1 summarizes both standard deviation and the variability (standard deviation / mean) to demonstrate the impact of the different magnitudes of ozone that result from model bias on both the absolute standard deviation (ppbv) and the relative standard deviation as represented by variability (%). We have also added a clarification:

Lines 305-307: "*We include this relative standard deviation metric since the CAM-chem biases make it difficult to compare standard deviations directly.*"

line 283 – Add "(Figure 2, Table 1)" after "Here".

This has been added, Lines 300-301.

lines 283-285 – This sentence is repetitive of the first paragraph in this section.

We have removed this sentence (and the insertion from the previous comment has been moved to the first paragraph of this section).

line 289 – Add "from continental to a single NE U.S. grid box" after "telescoping re-gions".

This has been added, Line 342.

line 290 – Add "albeit with lower overall variability" after "captures this trend".

This has been added, Line 343.

3.2 Variability, Averaging Windows, and Thresholds

line 314 – Add "underlying variability at the" before "particular choice of spatial and temporal scale".

This has been added, Line 358.

line 328 – Does "variability" here refer to standard deviation (as suggested by the ppbv thresholds) or as previously used, the relative variability (s.d./mean)? Confusing. Make sure to define the quantities being discussed.

We do not mean the previously defined definition of variability, so we have clarified this on Line 395, where we replaced "variability" with "anomaly for any selection of averaging window".

line 329 – Clarify what is meant here by "This difference".

This has been clarified on Line 395, replacing the word "difference" with "potential error."

3.3 Selection of Temporal Averaging Scales

line 358-359 – Add "meteorological variability causing ozone anomalies" before "exceeding particular thresholds", if this is the intended meaning.

This interpretation is the intended meaning, so "meteorological variability causing ozone anomalies" has been added to line 430.

line 363 – "Increas[ing] the threshold" is not really a strategy for "filtering out the noise". It is more like accepting the higher level of noise.

This has been clarified:

Line 435: "…either average over longer periods, or *acknowledge the level of noise and* increase the threshold."

lines 367 -370 – Confusing as written. Separate out the mention of Fig.S3 to a second sentence, e.g., "Similarly, in Supplemental Figure S3, one column (the 5-year averaging window) is selected."

We agree that these sentences were confusing as written. They have been updated and clarified:

Lines 439-442: "Supplemental Figure S3 *extends the analysis of Figure 5 by comparing* the MOZ_2000, MOZ_2050, and MOZ_2100 simulations *across the four thresholds for* the 5-year averaging window. *Figure 6 similarly compares* the 1 ppbv ozone *threshold*

*across the four averaging windows for* MOZ_2000, MOZ_2050, and MOZ_2100."

line 369 – "Figure 6" –> "Figure 5"

We have clarified this section, Lines 439-442.

 line 369 – Add "compare with" before "equivalent plots".

We have clarified this section, Lines 439-442.

line 370 – "Figures 7" –> "Figures 6".

We have clarified this section, Lines 439-442.

4. Discussion

line 434 – Add "variability" after "surface ozone".

We have added this text, Line 525.

line 460 – Cut comment in parentheses about future simulations. It is not known whether the future simulations will/would exhibit biases.

We agree with the reviewer, and have deleted this text.

5. Conclusions

line 502 – Add "and" after "configurations".

We have added this text, Line 603.

line 506 – Add "summertime" before "surface ozone". Clarify throughout conclusions that the analysis presented here is restricted to summer.

We have added the phrase "summertime" before references to ozone throughout the conclusion section (Lines 598, 607, 623, 628, and 645).

line 513 – Add "summertime" before "ozone variability".

We have added this text, Line 623.

line 523 – As mentioned earlier, the discussion of trend detection in the manuscript is very weak. Much more could (and should) be said about the application of the averaging methods presented here for trend detection. For instance, what are the implications of needing 10-15 year averaging windows for the length of timeseries needed to detect ozone trends (e.g., forced by climate change or emissions changes)?

In addition to additional examination of the implications of the 10 – 15 year averaging window ((Lines 43-46, Lines 611-615), we add the following text:

Lines 652-656: "*While we have detrended the CASTNET observations to compare to the constant year-2000 cycled emissions in the simulations, the CASTNET time series inherently includes the compounded variability of both meteorological and emission sources. Future studies will need to expand this analysis to include trends and variability in the emissions, as well as in the meteorology.*"

lines 524-530 – Mention here the compounding of (meteorological) variability in the observations with changes caused by variability/trends in emissions.

We address this along with the previous comment (Lines 633-644).

Figure 2 – Add the standard deviations plotted here standard deviations of daily ozone concentrations? If so, then for comparison with Figure 5, it would be useful also to show the interannual standard deviation of seasonal mean ozone.

These are for MDA8 $O_3$ mixing ratios, and is clarified in the caption (Line 899). Because the value of standard deviation would be different for every time and spatial scale, we don not think that it is practical to include interannual standard deviations here. We focus much of this manuscript on the variability and thresholds at the smallest spatial scales, which is represented in Figure 2 and Table 1.

Figure 3 – Explain that the CAM-chem simulation has fixed year-2000 emissions and SST, but time-varying meteorology. Why are the CASTNET values for 2000 "de-trended", instead of showing raw 2000 values? Change "MOZART" to "CAM-chem". In legend text in panel (a), also change "MOZART" to "CAM-chem".

Explanation added, terms updated. The detrending is centered at the year 2000, so the raw and detrended values are the same. This has been clarified in the caption, Lines 922-924.

Figure 4 – Define what is meant here by "variability". Is it the standard deviation, or the relative variability (s.d./mean)? Mention in caption that this plot shows summer ozone only. This is confusing from how the vertical axis is plotted.

It has been clarified that this is a plot of summertime MDA8 $O_3$ anomaly, Line 940.

Figure 8 – Change panel titles to the names of the regions. Keep the description of the regimes for filtering effectiveness in the text instead.

The panel titles have been updated in Figure 8 and the descriptions of the regions have been moved to the Caption of Figure 8 (Lines 980-982).

---

## Author Response (AR2)

Response to Reviewers for manuscript 'Maximizing Ozone Signals Among Chemical, Meteorological, and Climatological Variability'

Below we work through each of the reviewers' comments, with the comments in black and our responses in red. We also include any alterations to the text in red after our responses with the specific additions indicated with underlines.

Reviewer 1

The authors have addressed my main concerns satisfactorily with the improved Figure 2 and more careful caveating. I recommend the manuscript for publication with a few minor revisions/technical corrections, listed below.

Line 70: "Pawson" not "Dawson"

**Corrected, Line 76.**

Line 272-274: This sentence is rather convoluted. Please split in two or reword.

This sentence has been updated and clarified.

Lines 279-283: "Figure 2 compares summertime (JJA) maximum daily 8-hour average ozone (MDA8  $O_3$ ) from the present-day model simulation (MOZ 2000, Figure 2a) to the year-2000 CASTNET observations (Figure 2b). Figures 2c and 2d plot the MDA8  $O_3$  standard deviation and variability for MOZ\_2000, while Figures 2d and 2e compare the mean summertime MDA8  $O_3$  for the future simulations (MOZ 2050 and MOZ 2100)."

Line 288: But which is the most relevant metric, relative standard deviation or absolute? You later discuss thresholds in units of ppb, which suggests an absolute metric.

Both are relevant metrics that give a different view of the ozone comparisons. We have clarified the sentence to make this clearer.

Lines 301-304: "We consider both the standard deviation (ppb) and a mean-normalized standard deviation (as a percentage). The normalized standard deviation allows for a more direct comparison of the shape of the MDA8  $O_3$  distributions between the simulations and available observations, which accounts for the noted ozone biases (Figures 2b,c and Table 1)."

Reviewer 2

**General Comments**

The revisions to this manuscript addressed many of the concerns expressed in my original review, and helped to improve significantly the quality of the manuscript. The framework of a "sliding-scale" view of ozone variability is a particularly innovative new contribution from this

work. However, I have a few remaining concerns about the manuscript, which I list below. These concerns could be addressed through minor revisions and would not require new simulations or new substantive analyses.

Define better what is meant by "ozone signals". The intended meaning of this term is never defined. Throughout most of the paper, the emphasis is on the use of spatial and temporal averaging to decrease the meteorological variability. In this case, a robust estimate of the mean ozone mixing ratio (averaged appropriately) is the 'signal' that can be detected within a specified error tolerance. While this is one important type of ozone signal, the Abstract and Introduction also discuss ozone change 'signals', resulting for instance from emissions change or climate change, as opposed to meteorological variability. This type of ozone 'signal' is not discussed adequately in the paper. For instance, how can a forced trend in ozone concentrations be detected? Given the recommendations of 10-15 year averaging period, would this suggest the need for a 10-15 year average before an assumed instantaneous change in emissions, followed by a 10-15 year average afterwards? Presumably, the magnitude of the shift relative to the magnitude of the internal meteorological variability would be relevant to the required averaging period. Some discussion of such issues would greatly improve the usefulness and applicability of the methodologies in this paper.

We have added the following two paragraph to the end of the discussion section to address this point:

Lines 546-576: "Smaller signals require longer temporal averaging periods to identify. Figure 4 shows that a 0.5 ppb MDA8  $O_3$  signal will emerge after 15 - 20 years of temporal averaging. The range here reflects different spatial averaging domains, with larger domains requiring shorter temporal averaging windows than smaller domains (i.e. 15 years for averaging over the Continental US and 20 years for averaging over the Northeastern US). This would mean that an average trend of 0.25 - 0.33 ppb/year would require a time series of at least 15 years to identify. Similarly, a 1.0 ppb MDA8 O3 signal emerges after 7 - 15 years, which indicates an average trend of 0.14 -0.67 ppb/year would take at least 7 years to identify. Finally, a 5 ppb signal can be identified in less than 3 years, which indicates that an average trend of 1.67 ppb/year or greater would only require a 3-year time series. This presents particular difficulties if the ozone signal of interest is a trend spanning a time period on the same order. The 10 - 15 year averaging time scale we propose translates into a length of time beyond which you are likely to not see spurious trends above 0.5 ppb, but there are many cases in which the identification of a small trend is desired with less than 10 - 15 years of available data. For instance, Jiang et al. (2018) have found that NOx emissions reductions since 2005 are not as strong as previously expected, showing a significant slowdown beginning in 2011. This has large implications for ozone and for short-term decisions for air quality managers within the United States, who have to promulgate policies on short-term scales without the luxury of postponing action until longer and more complete data sets become available. As we have shown, spatial and temporal variability due to meteorology is high, and the identification and quantification of trends over 5, 10, or 15 years is difficult, particularly at small spatial scales.

*However, as we have shown, a consideration of the impact on variability – and how variability changes over time – is often pivotal to understand the nature of the signals being examined. In*

this paper, we have provided methods for quantifying the spatial and temporal variability and strategies for determining which types of signals are likely detectable at particular temporal and spatial scales. Some signals, especially small signals at small scales, are simply not large enough to emerge from the variability, and thus may not be detectable without additional data or expanding the temporal and spatial averaging scales used for analysis. Quantifying the signalto-noise ratio at a variety of spatial scales, and determining an acceptable threshold of a particular signal, could be one accessible method for providing this context. The risk in neglecting the quantification and contextualization of the magnitude of the ozone signal relative to the magnitude of the variability induced by the internal meteorology – and the impact of temporal and spatial averaging – is primarily the risk of drawing conclusions that are more sensitive to a particular peculiarity in the underlying variability rather than the signal itself."

We've also added the Jiang et al., 2018 reference to the references.

**Minor Comments**

Abstract, lines 43-46 -- Another possible reason for considering a shorter averaging period might be if a particularly large change (e.g., from rapid emissions reductions) were detected. If this change were large relative to the meteorological variability, a shorter averaging window could be used.

We have modified the abstract to make this point:

Lines 42-46: "For signals that are large compared to the meteorological variability (e.g. strong emissions reductions), shorter averaging periods and smaller spatial averaging regions may be sufficient, but for many signals that are smaller than or comparable in magnitude to the underlying meteorological variability, we recommend temporal averaging of 10 - 15 years combine with some level of spatial averaging (up to several hundred kilometers)."

We have also added a line to the conclusion:

Lines 620-622: "*For signals that are large compared to the underlying meteorological variability (e.g. strong emissions reductions), shorter averaging windows and smaller spatial regions may be used.*"

Section 2.1, lines 193-195 -- Sentence fragment. Revise.

This has been corrected:

Lines 200 – 202: "Offline forced meteorology *is taken* from the Modern-Era Retrospective analysis for Research and Applications (MERRA) reanalysis product..."

Section 4, line 447 -- "three" --> "four"

Corrected (Line 465).

Section 5, lines 538-540 -- Clarify what is meant by "the resultant magnitude of these changes" and "these changes and signals".

The sentence has been clarified:

Lines 591-592: "*However, these ozone signals (e.g. temporal trends or regional averages) are frequently small when* compared to the magnitude of the day-to-day ozone variability, and thus detecting these signals can be challenging."

Section 5, line 551 -- "out" --> "our"

Corrected (Line 604).

Section 5, line 558 -- Clarify the timescale over which the ozone variations of +/- 10-20 ppbv occur. Interannual variability of summer mean? Clarified:

Lines 615-616: "...summertime MDA8 O3 variability is largest at the smallest spatial and temporal scales..."